# Hierarchical Demonstration Order Optimization for Many-shot In-Context Learning

## Abstract

In-Context Learning (ICL) is a technique where large language models (LLMs) leverage multiple demonstrations (i.e., examples) to perform tasks. With the recent expansion of LLM context windows, many-shot ICL (generally with more than 50 demonstrations) can lead to significant performance improvements on a variety of language tasks such as text classification and question answering. Nevertheless, ICL faces demonstration order instability (ICL-DOI), which means that performance varies significantly depending on the order of demonstrations. Moreover, the ICL-DOI phenomenon persists and can sometimes be more pronounced in many-shot ICL, validated by our thorough experimental investigation. Current strategies handling ICL-DOI, however, are not applicable to many-shot ICL, since they cannot overcome two critical challenges: (1) Most metrics measuring the quality of demonstration order rely on subjective judgment, lacking a theoretical foundation to achieve precise quality characterization. These metrics are thus non-applicable to many-shot situations, where the order quality of different orders is less distinguishable due to the limited ability of LLMs to exploit information in long input context. (2) The requirement to examine all orders is computationally infeasible due to the combinatorial complexity of the order space in many-shot ICL. To tackle the first challenge, we design a demonstration order evaluation metric based on information theory for measuring order quality, which effectively quantifies the usable information gain of a given demonstration order. To address the second challenge, we propose a hierarchical demonstration order optimization method named `HIDO` that enables a more refined exploration of the order space, achieving high ICL performance without the need to evaluate all possible orders. Extensive experiments on multiple LLMs and real-world datasets demonstrate that our HIDO method consistently and efficiently outperforms other baselines. Our code can be found at `https://anonymous.4open.science/r/HIDO-B2DE/`.

## 1 Introduction

Large language models (LLMs) have demonstrated remarkable performance in few-shot In-Context Learning (ICL), i.e., adapting to new tasks or situations by utilizing demonstrations (examples) in the input prompt without additional training or fine-tuning (Brown et al., 2020; Dong et al., 2022; Zhao et al., 2023). Recent research advancements have enabled the deployment of LLMs with vastly expanded context windows, paving the way for many-shot ICL (Agarwal et al., 2024; Jiang et al., 2024; Li et al., 2023a; Bertsch et al., 2024; Moayedpour et al., 2024). This approach, typically involving more than 50 demonstrations, has achieved significant performance gains across various NLP tasks, including text classification (Min et al., 2022) and question answering (Li et al., 2023b). However, a critical challenge in few-shot ICL is *demonstration order instability* (ICL-DOI), which refers to the significant performance variance of ICL when the same set of demonstrations is arranged in different orders (Lu et al., 2022). For instance, Lu et al. (2022) claims that for a text classification task, different orders can cause performance to fluctuate dramatically, ranging from 90% accuracy to random guessing. Unfortunately, through exploratory experiments shown in Fig. 1 (see complete results in Appendix C.2), we observe that the ICL-DOI phenomenon persists in many-shot ICL scenarios and can be even more pronounced than in few-shot situations.

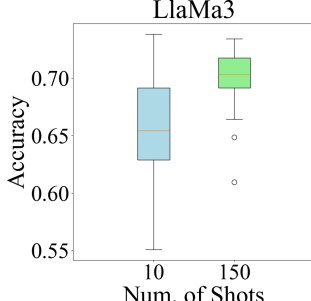 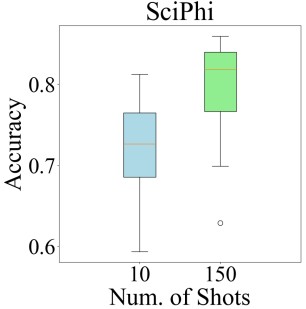 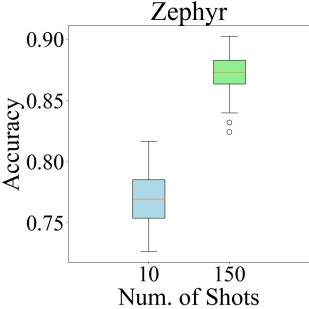

Figure 1: Accuracy difference between many-shot ICL performance (150 shots) and few-shot ICL (10 shots) on TREC. We randomly select different demonstration orders and test against 256 queries to determine the average times the model predicts the correct answer. This figure shows that ICL performance variance w.r.t. demonstration orders remain significant under many-shot scenarios.

Several studies tackle the issue of ICL-DOI in few-shot scenarios. One thread of research design stabilization methods to lower performance variance of ICL with different demonstration orders (Chen et al., 2023; Zhang et al., 2024; Xiang et al., 2024), while others search for the optimal demonstration orders such that the LLM achieves the highest prediction accuracy for the ICL task (Lu et al., 2022; Xu et al., 2024; Liu et al., 2024b). Although these proposed methods achieve satisfying performance under few-shot ICL, they can hardly be adapted to many-shot scenarios (Agarwal et al., 2024) due to two fundamental challenges: (1) **Lack of precise quality-measuring metric for demonstration order:** Existing research relies on subjective judgments when designing heuristic metrics for evaluating demonstration order quality. Thus, these metrics lack a theoretical foundation and can be noisy. However, LLMs are prone to pay more attention to the content in the beginning and the end (known as exhibit primacy bias and recency bias) in large context windows (Liu et al., 2024a). Therefore, for a large number of demonstrations, if the relevant demonstration is in the middle of the context, it would be difficult to distinguish the better order as the LLM may output results with subtle performance gap. (2) **Infeasibility of evaluating all demonstration orders:** Unlike few-shot ICL, where the existing demonstration order optimization methods evaluate every possible demonstration order, it is infeasible to conduct exhaustive demonstration order evaluations in many-shot scenarios. This is because evaluating one demonstration order requires at least one inference call, which is both costly and time-consuming. Meanwhile, the demonstration order space expands super-exponentially ($n!$) with the increase of demonstration numbers.

In this paper, we aim to take the initial step to address the issue of ICL-DOI in many-shot ICL by searching for an effective demonstration order. Specifically, to tackle the first challenge, we introduce the In-Context Demonstration Order $V$-information (ICD-OVI) score. This metric, grounded in information theory, measures how effectively an LLM, with a certain ordered demonstration as context, extracts usable information from a query to infer its corresponding answer. This metric measures the expected usable information that an ordered demonstration provides, which is interpretable and can utilize the information of test samples, computationally viable, and proved effective with extensive experiments. To address the second challenge, we introduce a **HI**erarchical **D**emonstration **O**rder optimization (`HIDO`) framework that enables more refined exploration in the order space thus achieving satisfactory ICL performance without evaluating all possible demonstration orders.

We summarize our contributions as follows: (1) **A Novel Metric with Theoretical Justification**: We introduce a novel score function ICD-OVI based on information theory for evaluating demonstration orders which is able to utilize the information from the probing set. (2) **A Fundamental Optimization Framework**: We propose a hierarchical demonstration order optimization framework termed `HIDO` for many-shot learning with vast demonstration permutation spaces. (3) **Extensive Empirical Evaluations**: We conduct extensive experiments on multiple LLMs and real-world datasets, demonstrating the effectiveness and efficiency of our `HIDO`.

## 2 PRELIMINARIES AND PROBLEM DEFINITION

*Notations*. Without further specification, we denote a demonstration as $d := (q, a)$, where $q$ is its query and $a$ is its answer. For example, a demonstration for sentiment classification can be

in the form of $(q, a) =$ ("My paper is accepted to ICLR2025!", "positive"). To transform the query-answer pair into a pure-text version suitable for LLM input, we apply a transformation $\mathcal{T}$. This transformation organizes the pair into a standardized text format using the following template: $\mathcal{T}(q, a) =$ "input: "$q$, "type: "$a$. By denoting the transformed text of the $i$th in-context demonstration $(q_i, a_i)$ as $\mathcal{T}_i = \mathcal{T}(q_i, a_i)$, the demonstration set to be ordered can be written as $\mathcal{D} := \{\mathcal{T}_i\}_{i=1}^{n}$ ($n$ is the number of demonstrations). An order permutation function, denoted as $\pi$, is defined as a bijective mapping from the set $\{1, ..., n\}$ to itself. We use $\Pi(\mathcal{D})$ to represent the text of concatenated demonstrations, ordered according to the permutation $\pi$, i.e., $\Pi(\mathcal{D}) := \mathcal{T}_{\pi(1)} \oplus ... \oplus \mathcal{T}_{\pi(n)}$, where $\oplus$ represents text concatenation operation.

***Preliminaries***. The ICL-DOI phenomenon was first proposed by Lu et al. (2022), who then developed two demonstration order evaluation metrics, GlobalE and LocalE, to assess the quality of a demonstration order given a set of LLM-generated probing samples $\hat{\mathcal{D}} := \{(\hat{q}_i, \hat{a}_i)\}_{i=1}^{T}$. Specifically, given ordered demonstrations $\Pi(\mathcal{D})$, GlobalE evaluate it with GlobalE$(\Pi(\mathcal{D})) = -\Sigma_i \mathbf{f}_i \log \mathbf{f}_i$. Here, the $\mathbf{f}$ is the LLM prediction label frequency vector, i.e., $\mathbf{f} = \frac{\Sigma_i \mathbb{I}[\arg\max P_{\text{LLM}}^{\Pi,i}(a)]}{T}$, where $P_{\text{LLM}}^{\Pi,i}(a) := P_{\text{LLM}}(a|\Pi(\mathcal{D}) \oplus \hat{q}_i)$ denotes the output distribution (i.e., logits vector) of the LLM, $\mathbb{I}(\cdot)$ the indicator function transforming an integer to its corresponding one-hot vector with length equal to the number of possible labels. GlobalE measures the diversity of labels given by the LLM under various probing samples. Lu et al. (2022) claim that label diversity maintains a high positive correlation with the accuracy of LLM predictions empirically. Therefore, demonstrations with higher GlobalE values are considered preferable.

Additionally, LocalE is calculated as the average entropy of LLM prediction (i.e., logits vector) on probing sets, i.e., LocalE$(\Pi(\mathcal{D})) = \frac{1}{T} \left[ \Sigma_i \Sigma_a P_{\text{LLM}}^{\Pi,i}(a) \log P_{\text{LLM}}^{\Pi,i}(a) \right]$. Unlike GlobalE, which measures the label frequency distribution across probing samples, LocalE focuses on the average uncertainty of the model's predictions for individual samples. Higher LocalE values indicate that the model has less confidence in its predictions, which helps prevent the LLM from being overconfident and poorly calibrated. However, GlobalE and LocalE are heuristic metrics inspired by their empirical observations and do not utilize the label information of the probing samples as they are not able to verify the correctness of those labels.

Another existing demonstration order quality metric is probability distribution optimization (PDO) metric (Xu et al., 2024) defined as PDO $= D_{\text{KL}} \left( \frac{1}{T} \Sigma_i P_{\text{LLM}}^{\Pi,i}(a) || U_A \right)$, in which $U_A$ is the uniform probability distribution of the label space. This metric aims to minimize the prediction label distribution discrepancy produced by LLM and the prior distribution (i.e., uniform distribution), which is guided under their assumption that well-ordered in-context examples should produce label distributions matching the prior label distribution. Nevertheless, the prior distribution of sample labels is not necessarily uniform, which has led to debates about its effectiveness and generalizability.

***Problem Definition***. Here, we formulate the in-context learning demonstration order optimization task as finding the order that minimizes the distribution discrepancy between the LLM output and the original input. Specifically, we have the following definition:

**Definition 1.** *For a demonstration data distribution $P(\cdot)$, where each data sample are in the shape of (query, answer), given $n$ demonstrations i.i.d. drawn from $P$, denoted as $\mathcal{D}$, we aim to find the demonstration order $\hat{\pi}$ of the $n$ i.i.d. samples such that the label prediction distribution produced by LLM approximates $P$, i.e.,*

$$\hat{\pi} = \min_{\Pi} KL(P_{LLM}(a|\Pi(\mathcal{D}) \oplus q)||P(a|q)). \tag{1}$$

## 3 IN-CONTEXT DEMONSTRATION ORDER $V$-USABLE INFORMATION

Before introducing our proposed HIDO model, we first present a novel evaluation metric termed In-Context Demonstration Order $\mathcal{V}$-usable Information (ICD-OVI). Unlike traditional heuristic ICL demonstration order metrics such as GlobalE (Lu et al., 2022), LocalE (Lu et al., 2022), and PDO (Xu et al., 2024), our ICD-OVI is the first metric to evaluate the quality of an ICL demonstration order with a theoretical foundation built on information theory and is capable of using the label information from the probing samples, hence being data efficient.

The design of ICD-OVI is inspired by $V$-usable information (Xu et al., 2023; Lin et al., 2023), a widely recognized information-theoretic metric measuring the amount of information an ML model can capture from input queries random variable $Q$ to predict their corresponding labels random variable $A$. Specifically, for a predictive family $\mathcal{V}$ (i.e., possible set of a model's configurations), the $\mathcal{V}$-usable information is defined as $H_\mathcal{V}(A) - H_\mathcal{V}(A|Q)$, where

$$H_\mathcal{V}(A|Q) = \inf_{f \in \mathcal{V}} \mathbb{E}_{(q,a) \sim P}[-\log f[q](a)],$$
$$H_\mathcal{V}(A|\emptyset) = \inf_{f \in \mathcal{V}} \mathbb{E}_{(q,a) \sim P}[-\log f[\emptyset](a)]. \tag{2}$$

Here, $P$ is the input data distribution, $f[q](a)$ is the predicted answer distribution given the information received from the query $q$. This metric has been shown to have multiple advantages: (1) **Interpretable**: This metric measures the amount of information (in units of "bits") of $Q$ that a model with predictive family $\mathcal{V}$ can capture to predict $A$, which is easily human-comprehensive. (2) **Computationally Viable**: Although the data distribution $\mathcal{D}$ is not accessible, it can be efficiently approximated by Monte Carlo with a theoretical precision guarantee (Xu et al., 2023). (2) **Empirically Effective**: the metric is empirically proven with a high correlation with the correctness of the predicted label (Lin et al., 2023; Yang et al., 2024; Wang et al., 2024).

Enlightened by $\mathcal{V}$-usable information, our ICD-OVI, measures the usable information that an LLM can capture from ordered demonstrations $\Pi(\mathcal{D})$. First, we define the predictive family corresponding to the ordered demonstrations $\Pi(\mathcal{D})$ as

$$\mathcal{V}_\Pi := \{P_{\text{LLM}}(\cdot|\Pi(\mathcal{D}) \oplus q)|q \in \mathcal{Q}_P\} \cup \{P_{\text{LLM}}(\cdot|q)|q \in \mathcal{Q}_P\}, \tag{3}$$

where $\mathcal{Q}_P$ represents the set of all possible queries in the sample space of task distribution $P$, and $\{P_{\text{LLM}}(\cdot|q)|q \in \mathcal{Q}_P\}$ is added to satisfy the optimal ignorance requirement for a predictive family (Xu et al., 2023). Then, ICD-OVI, the information that the model can capture from $\Pi(\mathcal{D})$, can be defined as the expected information the model with predictive family $\mathcal{V}_\Pi$ can capture from query random variable $Q$ for predicting label random variable $A$, i.e.,

$$\begin{aligned} \text{ICD-OVI} &= H_{\mathcal{V}_\Pi}(A) - H_{\mathcal{V}_\Pi}(A|Q), \\ &= \inf_{f \in \mathcal{V}_\Pi} \mathbb{E}_{q,a \sim \mathcal{D}}[-\log f[\emptyset](a)] - \inf_{f \in \mathcal{V}_\Pi} \mathbb{E}_{q,a \sim \mathcal{D}}[-\log f[q](a)], \\ &= \mathbb{E}_{(q,a) \sim P}[\log_2 P_{\text{LLM}}(a|\Pi(\mathcal{A}) \oplus \emptyset) - \log_2 P_{\text{LLM}}(a|\Pi(\mathcal{D}) \oplus q)], \end{aligned} \tag{4}$$

where $\Pi(\mathcal{A}) := \bigoplus_{i=1}^n \mathcal{T}(\emptyset, a_{\pi(i)})$. The third equation follows the definition of in-context $\mathcal{V}$-information from Eq. 1 of Lu et al. (2023). Practically, denoting $P_{\text{LLM}}^i(\hat{a}) := P_{\text{LLM}}(\hat{a}|\hat{q}_i)$, we may approximate the Eq. 4 with the probing samples $\hat{D}$ generated by LLM with

$$\frac{1}{|\hat{D}|} \Sigma_i (-\log_2 P_{\text{LLM}}^{\Pi,i}(\hat{a}) + \log_2 P_{\text{LLM}}^i(\hat{a})). \tag{5}$$

Nevertheless, Eq. 5 involves the LLM-generated labels $\hat{a}$s for the probing samples, which can be factually incorrect. Utilizing those incorrect labels may lead to bias in the computation of ICD-OVI. Fortunately, the theory of $V$-usable information (Ethayarajh et al., 2022; Lu et al., 2023) provide a effective tool called point-wise $\mathcal{V}$-informationn threshold (*PVI threshold*) which assists deciding if one generated probing sample label is reliable. Here, PVI is defined as

$$\text{PVI}_{(\hat{q},\hat{a})}^{\Pi(\mathcal{D})} = -\log_2 P_{\text{LLM}}(\hat{a}|\Pi(\mathcal{D}) \oplus \hat{q}) + \log_2 P_{\text{LLM}}(\hat{a}|\Pi(\mathcal{A}) \oplus \hat{q}). \tag{6}$$

By Eq. 6, the ICD-OVI is the mean of PVIs for all probing samples $\hat{\mathcal{D}}$. Built upon PVI, the PVI threshold is a scalar characterizing the likelihood of the correctness of the sample label. Specifically, when the PVI of a probing sample $(\hat{q}, \hat{a})$ is smaller than $\tau$, the label $\hat{a}$ is possibly incorrect; otherwise, the label $\hat{a}$ is highly likely to be correct for query $\hat{q}$. The existence of a PVI threshold is extensively validated by Ethayarajh et al. (2022); Lu et al. (2023) in various datasets and LLMs.

With the aid of the PVI threshold, we can address the potential bias caused by incorrect LLM-generated labels. Specifically, for a probing sample $(\hat{q}, \hat{a})$, we first calculate its PVI; if it is higher than a predefined $\mathcal{V}$-information threshold $\tau$, then we adopt the PVI of the sample $(\hat{q}, \hat{a})$ into the ICD-OVI calculation of ordered demonstrations $\Pi(\mathcal{D})$. Otherwise, we relax the PVI to its expectation for labels set $\{a|a \in \mathcal{A}\}$, i.e.,

$$\text{EPVI}_{(\hat{q},\hat{a})}^{\Pi(\mathcal{D})} = \Sigma_{a \in \mathcal{A}}[-P_{\text{LLM}}^{\Pi,\hat{q}}(a)\log_2 P_{\text{LLM}}^{\Pi,\hat{q}}(a) + P_{\text{LLM}}^{\hat{q}}(a)\log_2 P_{\text{LLM}}^{\hat{q}}(a)]. \tag{7}$$

Conclusively, by denoting point-wise ICD-OVI (PICD-OVI) as

$$\text{PICD-OVI}_{(\hat{q},\hat{a})}^{\Pi(\mathcal{D})} = \mathbb{I}(\text{PVI}_{(\hat{q},\hat{a})} \geq \tau)\text{PVI}_{(\hat{q},\hat{a})} + \mathbb{I}(\text{PVI}_{(\hat{q},\hat{a})} < \tau)\text{EPVI}_{(\hat{q},\hat{a})}, \tag{8}$$

our ICD-OVI can be approximated as

$$\text{ICD-OVI}(\Pi(\mathcal{D})) \approx \frac{1}{|\hat{D}|}\Sigma_{(\hat{q},\hat{a})}\text{PICD-OVI}_{(\hat{q},\hat{a})}. \tag{9}$$

Thus, our proposed ICD-OVI can effectively estimate the $V$-usable information despite noisy labels. Specifically, we have the theorem:

**Theorem 1.** *Under mild condition, for any two ordered demonstrations $\Pi_1(\mathcal{D})$ and $\Pi_1(\mathcal{D})$, given a probing sample $(\hat{q}, \hat{a})$, if*

$$PICD\text{-}OVI_{(\hat{q},\hat{a})}^{\Pi_1(\mathcal{D})} > PICD\text{-}OVI_{(\hat{q},\hat{a})}^{\Pi_2(\mathcal{D})}, \tag{10}$$

*then we have*

$$PVI_{(\hat{q},a^*)}^{\Pi_1(\mathcal{D})} > PVI_{(\hat{q},a^*)}^{\Pi_2(\mathcal{D})}, \tag{11}$$

*where the $a^*$ is the ground-truth label corresponding to the generated query $\hat{q}$. Therefore, if $\Pi_1(\mathcal{D})$ is more performant demonstration order than $\Pi_2(\mathcal{D})$, i.e., Eq. 14 establish for any probing sample $(\hat{q}, \hat{a})$, then ICD-OVI$(\Pi_1(\mathcal{D})) > $ICD-OVI$(\Pi_2(\mathcal{D}))$.*

Notably, although each probing sample $(\hat{q}, \hat{a})$ appears to require two LLM inference calls (one inference call for $\Pi(\mathcal{D}) \oplus \hat{q}$, the other for $\Pi(\mathcal{A}) \oplus \emptyset$ in Eq. 9, we only need to calculate $P_{\text{LLM}}(\hat{a}|\Pi(\mathcal{A}) \oplus \emptyset)$ once for one demonstration order regardless of the choice of probing sample $(\hat{q}, \hat{a})$. This ensures that ICD-OVI has comparable computational complexity to traditional heuristic metrics. Our ICD-OVI is the first information-theoretic metric for ICL demonstration order evaluation and inherits all the benign properties of $\mathcal{V}$-information in the scenario of ICL-DOI. Extensive empirical validations in our section of experiments show the effectiveness of our proposed ICD-OVI.

## 4 METHODOLOGY

In this section, we first present the motivations behind the key characteristics of our HIDO model design. Then, we provide an overview of our HIDO framework, followed by a detailed elaboration on each component in the framework.

### 4.1 MOTIVATION OF MODEL DESIGN

As mentioned in the Section 1, simply evaluating all possible demonstration orders is infeasible. Thus, we adopt a clustering method to more effectively search the permutation space. In this case, we transform the ICL-DOI problem to a hierarchical optimization, which solely requires determining the optimal order of demonstrations within each cluster and the optimal inter-cluster orders. This procedure significantly restricts the permutation search space from $n!$ ($n$ is the number of demonstrations in $\mathcal{D}$) to $k! \left[\frac{n}{k}\right]!$ ($k$ is the number of clusters). For example, if $n = 15$, $k = 5$, then the search space size will decrease from $1.3 \times 10^{12}$ to 720. To perform the hierarchical optimization, we first optimize the demonstration orders within each cluster. Then, with the fixed optimal demonstration order within each cluster, we optimize the inter-cluster orders.

In both intra-cluster and inter-cluster order optimization, a crucial step is evaluating the optimized order with our proposed ICD-OVI metric. However, ICD-OVI requires a probing set generated from LLM, which should model the distribution of the input data. For data efficiency, we assume that the demonstration order optimized for answer prediction also works well for sample generation, reflecting the input data distribution effectively. This assumption is empirically justified in Appendix C.3 and Appendix C.5. Based on this assumption, we can optimize the estimation precision of ICD-OVI by generating higher-quality probing sets (i.e., reducing the discrepancy between the probing sample distribution and the task data distribution) with the current optimized order after each intra-cluster and inter-cluster iteration (see details in Section 4.6). The newly generated probing sets are then used to evaluate the next iteration's optimized order, but the estimated ICD-OVI become more precise due to the increased quality of the probing sets.

Figure 2: Overview of our proposed in-context demonstration order optimization framework HIDO.

## 4.2 HIDO OVERVIEW

Fig. 2 illustrates the workflow of our proposed HIDO framework. In summary, HIDO first clusters the embeddings of the input demonstration texts and then performs $k$ iterations of hierarchical order optimizations. In each iteration, the process first determines the near-optimal order within each cluster. Then, while maintaining these intra-cluster orders, it searches for the most effective order of the clusters themselves. This alternating focus on intra- and inter-cluster optimization may be iterated multiple rounds during which the probing samples are imporved (see detailed rationale in Section 4.6) to achieve more accurate assessment of the demonstration order quality using ICD-OVI.

## 4.3 DEMONSTRATION CLUSTERING

According to Section 4.1, clustering demonstrations would substantially reduce the permutation space, allowing for more efficient search of the best order. Additionally, embeddings within the same cluster would be closer together, meaning less variance between the intra-cluster demonstrations. Thus, we apply a $K$-means algorithm (Macqueen, 1967) to the text embeddings of the demonstrations. These text embeddings are generated using the text embeddings API from OpenAI (2024a). We limit the number of clusters to be small (typically no more than four), as a larger number would cause a combinatorial explosion during HIDO's inter-cluster order optimization stage, where all possible orders are evaluated.

## 4.4 INTRA-CLUSTER ORDER OPTIMIZATION

In Section 4.3, we restrict the cluster number to be small (typically no more than four) so that we can evaluate all the inter-cluster orders. This implies that demonstrations within one cluster, despite sharing similar latent embeddings, can be large in quantity. For instance, a typical many-shot in-context learning process requires between 50 and 150 demonstrations (Ye et al., 2023; Agarwal et al., 2024), so the number of samples within a cluster can be as large as 30, making it impossible to evaluate all their permutation combinations. Nevertheless, the intra-cluster demonstrations share proximate embeddings, which significantly decreases ICL performance variance when demonstration orders vary. This allows a less thorough order search while still achieving satisfactory precision.

Hence, we design a demonstration order space exploration strategy as follows (see the illustration in Fig. 3): we first randomly generate a demonstration order as the starting point. Then, in each iteration, we explore its "neighborhood" by randomly flipping 10% of its positions, which ensures sufficient variation between selected orders while constraining exploration within a defined radius of the anchor order, as measured by rank correlations. Specifically, we have (see proof in Appendix B)

**Theorem 2.** *Randomly flipping $K$ entries from a sequence of length $N$ will always keep the rank correlation within a range characterized by the lower bound $1 - 6\sum_{i=1}^{K}(a_i - a_{K+1-i})^2/N(N^2 - 1)$ and upper bound $1$. The upper bound is achieved with a extremely low probability of $1/K!$ when the perturbed sequence is identical to the original sequence.*

For each candidate intra-cluster order, we evaluate its quality using the ICD-OVI metric, which relies on a probing set generated by a language model (LLM). Here, the probing set generation process leverages information from the previous optimization iteration. Specifically, we start with the top $k$ effective intra-cluster orders for the cluster of interest from the previous optimization iteration. For each of these $k$ orders, we create $k$ distinct sets of ordered demonstrations by combining: *(i)* The *optimal inter-cluster order* from the previous iteration (fixed); *(ii)* The *optimal intra-cluster orders for **all other** clusters* from the previous iteration (fixed); *(iii)* The

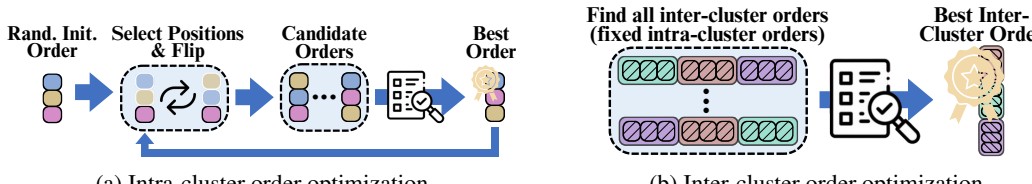

(a) Intra-cluster order optimization        (b) Inter-cluster order optimization

Figure 3: Illustrations of intra-cluster and inter-cluster order optimization

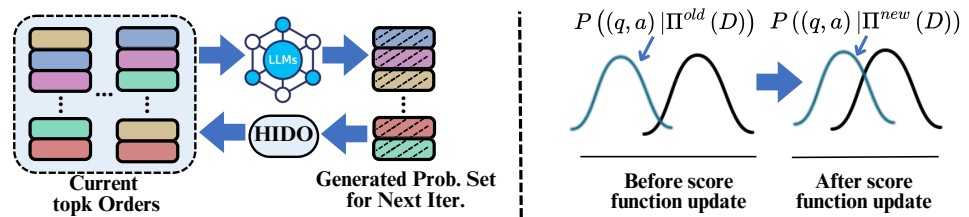

Figure 4: Illustration for dynamic update of the score function.

*current candidate intra-cluster order* (from the $k$ orders) for the cluster being optimized. The $k$ distinct ordered demonstrations differ only in the order of demonstrations within the cluster of interest. We then prompt the LLM with each of the $k$ sets, generating $k$ different probing sample sets. These $k$ probing sets are collectively used to evaluate the quality of the candidate intra-cluster order with the ICD-OVI metric. Using multiple probing sets derived from the top performing orders of the previous iteration, we achieve a more robust and comprehensive candidate order evaluation.

## 4.5 INTER-CLUSTER ORDER OPTIMIZATION

Having obtained the near-optimal demonstration orders within each cluster, we now focus on finding the optimal order of the clusters themselves. As we have limited the number of clusters to typically no more than four, it becomes feasible to evaluate all possible cluster orders, as illustrated in Fig. 3 (b). Similar to the intra-cluster optimization process, we generate a probing set to evaluate each possible inter-cluster order. However, in this case, we employ all possible cluster orders, while fixing the optimal intra-cluster demonstration orders obtained from the previous iteration.

Specifically, we first consider all possible permutations of cluster orders, then prompt the LLM with this complete set of ordered demonstrations (combining the cluster order being evaluated and the fixed optimal intra-cluster orders) to generate a probing set. Each generated probing set is used to evaluate its corresponding inter-cluster order using the ICD-OVI metric. This approach allows us to comprehensively assess different cluster arrangements while leveraging the optimized intra-cluster orders, potentially leading to a globally optimized demonstration order.

## 4.6 DYNAMIC UPDATE OF THE SCORE FUNCTION

Our `HIDO` performs multiple rounds of intra- and inter-cluster optimization, during which the score function (ICD-OVI evaluation) is refined through updated probing sets, which is illustrated in Fig. 4. A higher-quality probing set reduces distribution discrepancy between probing and input data samples, enabling more precise ICD-OVI estimation. This procedure further improves accuracy in identifying effective demonstration orders for answer prediction.

This procedure is separately introduced in the Section 4.4 and Section 4.5, therefore, we briefly concluded it as follows. In each iteration of in-context demonstration order optimization, we cache the top $k$ intra-cluster demonstration orders for all clusters. For intra-cluster optimization in the subsequent iteration, we apply the cached top $k$ orders for the cluster being optimized, while maintaining the optimal intra-cluster orders from the previous iteration for all other clusters. We combine these with the optimal inter-cluster order from the previous iteration to generate new probing sets. For inter-cluster optimization, we consider all possible cluster arrangements. For each arrangement, we apply the optimal intra-cluster demonstration orders obtained from the previous iteration to generate probing sets for evaluating each inter-cluster arrangement.

Table 1: Metadata of the LLMs tested. "Lan. models", "Con. window" indicates the language models, and context window size.

| Lan. models | GPT-3.5T | GPT4oM | SciPhi | Zephyr | LlaMa3 |
|---|---|---|---|---|---|
| Con. window | 16,385 | 128,000 | 32,768 | 32,768 | 1,048,576 |
| Max output | 4,096 | 16,384 | n/a | n/a | n/a |
| Model size | 175B | n/a | 7B | 7B | 8B |

Table 2: The performance of our `HIDO` along with baselines on various datasets. The best performance for each dataset and model combination is bolded.

| | | AGNews | CB | CR | DBPedia | MPQA | MR | RTE | SST-5 | TREC |
|---|---|---|---|---|---|---|---|---|---|---|
| GPT-3.5T | GlobalE | 87.24 ± 0.60 | 46.43 ± 9.28 | 93.36 ± 0.39 | 95.70 ± 1.41 | 90.76 ± 0.81 | 93.62 ± 0.98 | **81.90 ± 0.90** | 54.56 ± 2.83 | 77.47 ± 6.56 |
| | LocalE | 89.06 ± 0.39 | 46.43 ± 9.28 | 93.10 ± 0.81 | 95.83 ± 1.37 | 89.97 ± 0.60 | 93.49 ± 1.19 | 80.86 ± 0.39 | 52.60 ± 4.77 | 78.65 ± 6.72 |
| | PDO | 89.32 ± 0.45 | 48.21 ± 7.78 | 93.23 ± 0.60 | 96.22 ± 1.80 | 89.97 ± 0.98 | 93.62 ± 0.98 | 80.60 ± 0.45 | 53.65 ± 3.52 | 76.69 ± 6.08 |
| | HIDO | **89.45 ± 0.39** | **51.19 ± 2.73** | **94.27 ± 0.23** | **97.92 ± 0.23** | **91.02 ± 0.78** | **94.27 ± 0.45** | **81.90 ± 0.60** | **54.95 ± 1.48** | **82.29 ± 1.63** |
| GPT-4oM | GlobalE | 83.07 ± 3.37 | 55.95 ± 1.03 | **93.36 ± 0.39** | 92.19 ± 2.17 | **87.50 ± 1.79** | 92.71 ± 0.45 | 85.16 ± 0.78 | 53.39 ± 2.48 | 83.33 ± 2.15 |
| | LocalE | 84.77 ± 0.78 | 55.95 ± 1.03 | **93.36 ± 0.68** | 92.19 ± 2.56 | 86.33 ± 3.73 | 92.32 ± 2.22 | 85.55 ± 1.35 | 53.26 ± 2.60 | 84.11 ± 1.76 |
| | PDO | 85.03 ± 2.00 | 55.36 ± 0.00 | 92.84 ± 0.60 | 92.19 ± 2.34 | 81.64 ± 1.41 | 92.84 ± 1.13 | 85.42 ± 1.63 | 52.86 ± 2.22 | 84.51 ± 2.60 |
| | HIDO | **85.81 ± 2.22** | **56.55 ± 1.03** | **93.36 ± 0.68** | **92.84 ± 0.81** | 86.85 ± 1.13 | **93.23 ± 0.60** | **86.33 ± 0.68** | **56.64 ± 3.20** | **86.59 ± 1.37** |
| SciPhi | GlobalE | 85.29 ± 0.81 | **92.26 ± 1.03** | 91.67 ± 0.60 | 96.09 ± 1.41 | 83.59 ± 0.68 | 93.88 ± 0.45 | 83.72 ± 2.39 | 54.69 ± 2.38 | 76.17 ± 7.32 |
| | LocalE | 86.59 ± 0.23 | **92.26 ± 1.03** | 92.32 ± 1.13 | 96.22 ± 0.81 | 85.16 ± 0.68 | 93.88 ± 0.23 | 83.98 ± 0.39 | 55.08 ± 1.70 | 76.69 ± 3.16 |
| | PDO | 86.07 ± 0.60 | **92.26 ± 1.03** | 91.02 ± 1.70 | 96.09 ± 1.41 | 84.77 ± 0.39 | 94.01 ± 0.23 | 83.72 ± 2.39 | 54.69 ± 2.38 | 76.17 ± 7.32 |
| | HIDO | **86.98 ± 0.45** | 90.48 ± 1.03 | **92.71 ± 0.60** | **96.88 ± 0.68** | **87.50 ± 0.78** | **94.27 ± 0.45** | **85.94 ± 0.78** | **57.16 ± 1.85** | **80.47 ± 0.78** |
| Zephyr | GlobalE | **89.71 ± 0.98** | 77.38 ± 5.15 | 93.23 ± 0.90 | 94.66 ± 2.00 | 86.07 ± 1.26 | 94.40 ± 0.45 | 82.16 ± 1.13 | 50.00 ± 0.68 | 84.38 ± 1.17 |
| | LocalE | 88.15 ± 0.23 | 73.21 ± 4.72 | 93.10 ± 1.13 | 96.22 ± 1.97 | 86.98 ± 0.60 | 94.66 ± 0.23 | **82.55 ± 0.81** | 48.18 ± 1.85 | 81.90 ± 4.30 |
| | PDO | 88.80 ± 0.81 | 77.38 ± 5.15 | 93.23 ± 0.90 | 94.66 ± 2.00 | 86.07 ± 0.60 | 93.10 ± 0.98 | 81.51 ± 1.93 | 50.00 ± 0.68 | 84.38 ± 1.17 |
| | HIDO | 89.32 ± 0.90 | **78.57 ± 1.79** | **94.01 ± 0.45** | **97.27 ± 0.68** | **87.76 ± 0.98** | **94.79 ± 0.60** | **82.55 ± 1.37** | **50.78 ± 2.07** | **86.46 ± 1.48** |
| LlaMa3 | GlobalE | 80.34 ± 4.95 | **94.64 ± 1.79** | 85.94 ± 3.20 | 93.49 ± 1.48 | 58.20 ± 2.34 | 92.84 ± 0.81 | 82.42 ± 0.39 | 39.19 ± 1.93 | 72.92 ± 1.48 |
| | LocalE | 83.72 ± 4.49 | 91.67 ± 2.06 | 85.68 ± 4.77 | 93.23 ± 0.23 | 54.82 ± 1.26 | 90.49 ± 0.81 | 82.81 ± 1.03 | 40.36 ± 4.21 | 73.18 ± 6.79 |
| | PDO | 77.73 ± 1.03 | **94.64 ± 1.79** | 85.16 ± 2.38 | 93.49 ± 1.48 | 52.21 ± 1.26 | 91.02 ± 1.35 | 82.94 ± 0.23 | 39.19 ± 1.93 | 72.92 ± 1.48 |
| | HIDO | **86.20 ± 2.29** | **94.64 ± 3.09** | **87.24 ± 2.39** | **94.27 ± 1.58** | **63.80 ± 7.64** | **93.49 ± 0.98** | **83.07 ± 0.98** | **40.62 ± 3.58** | **77.34 ± 3.73** |

This approach is based on the assumption in Section 4.1 that the optimal demonstration order is also effective for sample generation. With this, we can reuse the most effective orders found for label prediction from the previous iteration when generating high-quality probing samples in the current iteration, which significantly reduces computational costs. By iteratively refining our probing sets for both intra-cluster and inter-cluster optimizations, we aim to improve the accuracy of our order evaluations progressively, leading to better optimized orders over time.

## 5 EXPERIMENTS

In this section, we first introduce our experimental setup, including the datasets, baselines, and LLMs utilized. Then, we present the main results demonstrating the effectiveness of `HIDO` compared to the baseline methods. Finally, we conduct ablation studies to examine the utility of different components and perform parameter sensitivity analysis to test the robustness of our approach. In particular, we focus on answering the three research questions via extensive experiments: **RQ1:** How does `HIDO` perform compared to existing demonstration order optimization methods across different datasets and language models? **RQ2:** What is the impact of each key component in `HIDO` on its overall performance? **RQ3:** How sensitive is `HIDO` to its main hyperparameters?

### 5.1 EXPERIMENT SETUP

Here, we introduce the various settings for our experimental evaluation.

***Baselines***: (1) **GlobalE**: Randomly select 24 orders and measure the entropy of the frequency distribution of the prediction labels on probing datasets Lu et al. (2022); (2) **LocalE**: Analogously to Lu

et al. (2022), randomly select 24 demonstration orders and calculate the average entropy of their predicted logits given by LLM. (3) **Probability Distribution Ordering (PDO)**: Randomly sample 24 orders and calculate the KL divergence between the frequency distribution of the prediction labels on probing datasets and the uniform distribution (Xu et al., 2024).

***Datasets***: We adopt nine text classification datasets, namely AGNews (Zhang et al., 2015), CB (De Marneffe et al., 2019), CR (Hu and Liu, 2004), DBPedia (Zhang et al., 2015), MPQA (Wiebe et al., 2005), MR (Pang and Lee, 2005), RTE (Dagan et al., 2005), SST-5 (Socher et al., 2013) and TREC (Voorhees and Tice, 2000). Those datasets cover various semantic scenarios, including sentiment classification and textual entailment (see Appendix C.4 for demonstration examples). For evaluation, we sub-sample 256 instances from each dataset due to budget constraints.

***Large Language Models***: We adopt "GPT-3.5-Turbo-0125" (OpenAI, 2024b) and "GPT-4o-Mini-2024-07-19" (OpenAI, 2024c) from OpenAI, "SciPhi-Mistral-7B-32k" (Huggingface, 2024b), "Zephyr-7b-beta" (Huggingface, 2024c) and "LlaMa-3-8B-Instuct-Gradient-1048k" (Huggingface, 2024a) from HuggingFace. We select the OpenAI models due to their affordability and the HuggingFace models due to their large context windows.

## 5.2 EFFECTIVENESS OF HIDO

In this section, we aim to answer **RQ1**. In Table 2, we measure the accuracy of the output demonstration orders produced by HIDO and the baselines on various datasets and LLMs. We observe that HIDO achieves the highest prediction accuracy in most settings, proving the effectiveness of our framework. Notably, our method can achieve significant performance leads in GPT-3.5T on CB (51.19%), GPT-4oM on SST-5 (56.64%), SciPhi on TREC (80.47%), and LlaMa3 on MPQA (63.80%). Additionally, we make the following observations from Table 2: (1) **Model-agnostic**: HIDO achieves the best performance on both large and small LLMs, implying that our framework is model agnostic; it can be used on different models and find relatively high-performing orders. (2) **Low variance**: In general, HIDO has a smaller variation in performance on most dataset model combinations in contrast to that of the baselines, especially in GPT-3.5T on CB (2.73%), GPT-4oM on DBPedia (0.81%), and SciPhi on TREC (0.78%). This indicates that HIDO can consistently find the order that gives the best performance. (3) **Runner-up on non-optimal datasets**: In those cases that HIDO does not perform the best, the results are still comparable to the best-performing baseline.

## 5.3 ABLATION STUDY

In this subsection, we address **RQ2** by examining four variants of our HIDO model: (1) **HIDO-NC**: This variant tests the effectiveness of our clustering procedure by randomly assigning samples to clusters. For a fair comparison, we maintain the same number of clusters and demonstrations per cluster as in the original HIDO. (2) **HIDO-NIntra**: Instead of optimizing each intra-cluster demonstration order, this variant randomly selects demonstration orders within clusters while keeping all other components the same as HIDO. (3) **HIDO-NInter**: After the intra-cluster demonstration order optimization stage, this variant randomly selects an inter-cluster order as the optimal inter-cluster order. (4) **HIDO-ND**: This variant removes the dynamic update scheme for the score function. It outputs the best demonstration order after only one optimization iteration.

From Fig. 5a, we observe that removing each component causes performance degradation. More specifically, we have the following observations: (1) HIDO-NC has the largest difference, indicating that grouping the samples based on distance allows HIDO to find the best order while maintaining efficiency. (2) HIDO-ND has relatively small increase, which implies that HIDO is able to find the best order within a small number of optimization iterations. (3) HIDO-NInter and HIDO-NIntra have similar impacts on the performance. This highlights the significance of hierarchical optimization in finding the best order. Additional results can be found in Appendix C.1.

## 5.4 PARAMETER SENSITIVITY

In this subsection, we address **RQ3**. Although our model has numerous hyperparameters, we focus our analysis on two we consider most significant: the number of clusters $k$ and the maximum number of optimization iterations $l$. Fig. 5b illustrates our model's performance with varying $k$ and $l$ on the TREC and MPQA datasets using the Sciphi model. We observe that performance generally

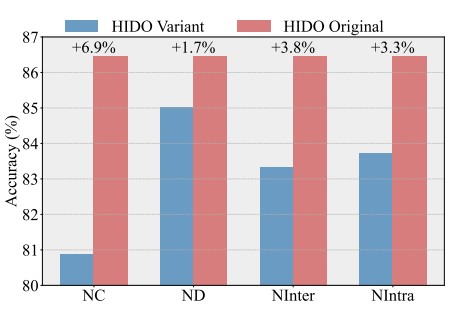

(a) Performance of `HIDO` and its variants using Zephyr on Trec.

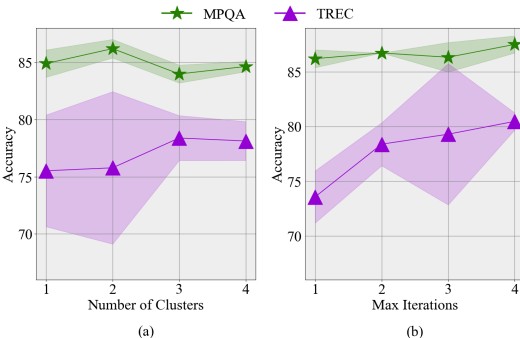

(b) Parameter analysis of `HIDO` using Sciphi on Trec.

Figure 5: Combined results of ablation study and parameter analysis.

improves as $l$ increases, indicating that more iterations of `HIDO` tend to produce better-performing demonstration orders. Regarding the number of clusters, we find that performance peaks at $k = 2$ for MPQA and $k = 3$ for TREC. This variation suggests that different datasets require specific numbers of clusters to optimally partition the data and yield the best-performing demonstration orders.

## 6 RELATED WORK

### 6.1 MANY-SHOT IN-CONTEXT LEARNING

With the expanded context window of recently developed LLMs, the models can process a larger number of demonstrations within a single prompt, resulting in further research observing the effect of large number of demonstrations (i.e. more than 50) on ICL (Agarwal et al., 2024; Jiang et al., 2024; Li et al., 2023a; Bertsch et al., 2024; Moayedpour et al., 2024). Li et al. (2023a) develop a long-range language model EVALM that achieves higher accuracy when using many shot ICL; however, the model cannot maintain the same performance consistently, indicating that ICL-DOI still exists. Some emprirical results from Agarwal et al. (2024) provides early evidence for many-shot demonstration order sensitivity by showing how one order that gives the best performance on one subset of a dataset can perform poorly on a different subset of the same original dataset.

### 6.2 OPTIMIZATION TECHNIQUES FOR VAST PERMUTATION SPACES

The problem of finding optimal orderings in large permutation spaces is not unique to ICL and has been studied in various domains. Traditional approaches like simulated annealing (Kirkpatrick et al., 1983) and genetic algorithms (Tomassini, 1995) have been applied to similar combinatorial optimization problems. However, these methods often struggle with the scale of permutations encountered in ICL scenarios. Recent work in combinatorial optimization has introduced hierarchical and decomposition-based approaches to tackle large-scale permutation problems (Goh et al., 2022; Luo et al., 2023; Pan et al., 2023). For instance, Pan et al. (2023) proposed a hierarchical optimization framework for solving large-scale traveling salesman problems, demonstrating the effectiveness of dividing the problem into manageable sub-problems. Enlightened by those ideas, we tackle specific challenges of ICL demonstration ordering.

## 7 CONCLUSION

This paper introduces `HIDO`, a novel approach to perform demonstration order optimization in in-context learning (ICL). `HIDO` efficiently navigates vast permutation spaces to find effective demonstration orders, significantly reducing search time while maintaining high prediction utility. Our key contributions include a score function with solid theoretical foundation based on information theory for evaluating demonstration orders, a hierarchical optimization framework, and a dynamic update mechanism. Extensive experiments on multiple LLMs and datasets demonstrate that our `HIDO` outperforms existing baselines. We hope that our work will shed light on new promising methods for unleashing in-context learning performance.

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

## A    LIMITATIONS

In this work, several limitations exist that should be acknowledged for a balanced understanding of the results and methodology. First, while the Hierarchical Demonstration Order Optimization (HIDO) framework effectively reduces the search space for many-shot in-context learning (ICL), its reliance on clustering introduces an additional layer of complexity that may not always generalize well to all datasets or language models. The clustering process itself, especially with a limited number of clusters, may not capture intricate interdependencies between demonstrations. Furthermore, although the dynamic update mechanism improves the accuracy of the score function, it also increases the overall computational cost, particularly when applied to very large datasets or when running a high number of optimization iterations.

Additionally, the current framework assumes that performance improvements arise primarily from the optimized demonstration order, but factors such as the inherent instability of large language models (LLMs) across varying contexts might also contribute to observed fluctuations. Finally, the probing set generation step introduces potential noise, and while the system attempts to mitigate this through iterative updates, inaccuracies in probing may still affect the final demonstration order selection.

## B    THEOREMS AND PROOFS

**Lemma 1.** *Let $f(x_1, \ldots, x_n) = \sum_{i=1}^{n} x_i \log x_i$ be defined for $x_i > 0$, with the constraint $\sum_{i=1}^{n} x_i = c$, where $0 < c < \frac{1}{e}$. Then:*

*1. $f$ reaches its minimum when all $x_i$ are equal, i.e., $x_i = \frac{c}{n}$ for all $i$.*

*2. $f$ reaches its maximum when one $x_i$ equals $c$ and the rest are zero.*

*Proof.* We will use the method of Lagrange multipliers.

Let $g(x_1, \ldots, x_n) = \sum_{i=1}^{n} x_i - c = 0$ be our constraint. The Lagrangian is:

$$L(x_1, \ldots, x_n, \lambda) = \sum_{i=1}^{n} x_i \log x_i - \lambda(\sum_{i=1}^{n} x_i - c)$$

We set the partial derivatives to zero:

$$\frac{\partial L}{\partial x_i} = \log x_i + 1 - \lambda = 0 \quad \text{for } i = 1, \ldots, n$$

$$\frac{\partial L}{\partial \lambda} = \sum_{i=1}^{n} x_i - c = 0$$

From $\frac{\partial L}{\partial x_i} = 0$, we get:

$$x_i = e^{\lambda - 1}$$

This shows that all $x_i$ are equal at the critical points.

**Minimum Point:** When all $x_i$ are equal, let $x_i = \frac{c}{n}$ for all $i$. The function value is:

$$f(\frac{c}{n}, \ldots, \frac{c}{n}) = c \log \frac{c}{n}$$

**Maximum Point:** Consider $x_1 = c$ and $x_i = 0$ for $i > 1$. The function value is:

$$f(c, 0, \ldots, 0) = c \log c$$

To show that $f(\frac{c}{n}, \ldots, \frac{c}{n}) < f(c, 0, \ldots, 0)$, we need to prove:

$$c \log \frac{c}{n} < c \log c$$

This is equivalent to $frac{c}{n} < c$, which is true for $n > 1$ and $c > 0$. Therefore, we have shown that the minimum occurs when all $x_i = \frac{c}{n}$, and the maximum occurs when one $x_i = c$ and the rest are zero. □

**Theorem 1** *We assume that given a LLM, a probing sample $(\hat{q}, \hat{a})$ and an ordered demonstration text $\Pi(\mathcal{D})$,*

- *When $PVI_{(\hat{q},\hat{a})}^{\Pi(\mathcal{D})} \geq \tau$, then $\hat{a} = a^*$, where the $a^*$ is the ground-truth label corresponding to the generated query $\hat{q}$.*

- *The LLM predict the label $\hat{a}$ with the highest probability when query by $\hat{q}$ with $\Pi(\mathcal{D})$ as its context, i.e., $P(\hat{a}|\Pi(\mathcal{D}) \oplus \hat{q}) = \arg\max_{a \in \mathcal{A}} P(a|\Pi(\mathcal{D}) \oplus \hat{q})$.*

- *Assume that for any two ordered demonstration texts $\Pi_1(\mathcal{D})$ and $\Pi_2(\mathcal{D})$, the $P_{LLM}(a|\Pi_1(\mathcal{A}) \oplus \emptyset) = P_{LLM}(a|\Pi_2(\mathcal{A}) \oplus \emptyset)$ for all $a \in \mathcal{A}$.*

*Without loss of generalizability, for any two ordered demonstrations $\Pi_1(\mathcal{D})$ and $\Pi_1(\mathcal{D})$, there is a $\epsilon(\frac{1}{e} \leq \epsilon \leq 1)$ such that $P(\hat{a}|\Pi_i(\mathcal{D}) \oplus \hat{q}) > 1 - \epsilon$. We additionally assume that when $PVI_{(\hat{q},\hat{a})}^{\Pi(\mathcal{D})} < \tau$:*

- *The $a^*$ is the second most probable label given by the LLM when prompted by query $\hat{q}$ with any ordered demonstration context $\Pi(\mathcal{D})$, i.e., $P(a^*|\Pi(\mathcal{D}) \oplus \hat{q}) = \arg\max_{a \in \mathcal{A} \setminus \{\hat{a}\}} P(a|\Pi(\mathcal{D}) \oplus \hat{q})$; we write $P(a^*|\Pi_i(\mathcal{D}) \oplus \hat{q}) = \lambda_i \epsilon$, where $0 \leq \lambda_i \leq 1$, $i \in \{1, 2\}$.*

- *By symmetry, we only consider the case $\lambda_1 < \lambda_2$. In this case, we assume that $\frac{1}{2} - \delta < \lambda_1 < \frac{1}{2} + \delta$ ($\delta$ is a constant) such that*

$$(\lambda_1 \epsilon) \log \lambda_1 \epsilon + (1 - \lambda_1 \epsilon) \log (1 - \lambda_1 \epsilon) < \epsilon \log \epsilon - (2 - \lambda_1)\epsilon. \tag{12}$$

*Meanwhile, we require $\lambda_2 - \lambda_1 > (1 - \frac{1}{\log(n-2)})(1 - \lambda_1)$.*

*With the assumptions above, if*

$$PICD\text{-}OVI_{(\hat{q},\hat{a})}^{\Pi_1(\mathcal{D})} > PICD\text{-}OVI_{(\hat{q},\hat{a})}^{\Pi_2(\mathcal{D})}, \tag{13}$$

*then we have*

$$PVI_{(\hat{q},a^*)}^{\Pi_1(\mathcal{D})} > PVI_{(\hat{q},a^*)}^{\Pi_2(\mathcal{D})}. \tag{14}$$

*Therefore, if $\Pi_1(\mathcal{D})$ is more performant demonstration order than $\Pi_2(\mathcal{D})$, i.e., Eq. 14 establish for any probing sample $(\hat{q}, \hat{a})$, then*

$$ICD\text{-}OVI(\Pi_1(\mathcal{D})) > ICD\text{-}OVI(\Pi_2(\mathcal{D})). \tag{15}$$

*Proof.* First, in the case that $PVI_{(\hat{q},\hat{a})}^{\Pi(\mathcal{D})} \geq \tau$, by Assumption 1, we have $\hat{a} = a^*$. Therefore, we have

$$\text{PICD-OVI}_{\hat{q},\hat{a}}^{\Pi(\mathcal{D})} = P(\hat{a}|\Pi(\mathcal{D}) \oplus \hat{q}) - P(\hat{a}|\Pi(\mathcal{A}) \oplus \emptyset) = \text{PVI}_{\hat{q},\hat{a}}^{\Pi(\mathcal{D})} = \text{PVI}_{\hat{q},a^*}^{\Pi(\mathcal{D})}. \tag{16}$$

Eq. 16 enforces the establishment of Eq. 14.

Next, in the case where $\text{PVI}_{(\hat{q},\hat{a})}^{\Pi(\mathcal{D})} < \tau$, with Assumption 3, it suffices to prove that $|\lambda_1 \epsilon \log \lambda_1 \epsilon| \geq |\lambda_2 \epsilon \log \lambda_2 \epsilon|$ gives rise to

$$|\lambda_1 \epsilon \log \lambda_1 \epsilon + \Sigma_{\Sigma_i x_i = (1-\lambda_1)\epsilon} x_i \log x_i + x_{\hat{a},1}| \geq |\lambda_2 \epsilon \log \lambda_2 \epsilon + \Sigma_{\Sigma_i x_i = (1-\lambda_1)\epsilon} x_i \log x_i + x_{\hat{a},2}|. \tag{17}$$

Now, by utilizing the Assumption 5, we claim that Eq. 17 establish, thus the theorem is proved.

To prove Eq. 17, we start from the known inequivality

$$\lambda_2 - \lambda_1 > (1 - \frac{1}{\log(n-2)})(1 - \lambda_1). \tag{18}$$

For simplicity, we represent $\lambda_2 - \lambda_1$ as $\Delta$ in the following texts. We rewrite the Eq. 18 as

$$\Delta > \frac{1 - 1/\epsilon \log e^{-\epsilon(1-\lambda_1) - \epsilon + \log_2}/2}{\log(n-2)} + (1 - \lambda_1),$$

$$= \frac{1}{\epsilon} \left[ \frac{\epsilon(\log \epsilon - \log 2) + (\log 2 - \epsilon(2 - \lambda_1))}{\log(n-2)} \right] - \frac{\log \epsilon}{\log(n-2)} + (1 - \lambda_1) + \frac{1}{\log(n-2)}. \tag{19}$$

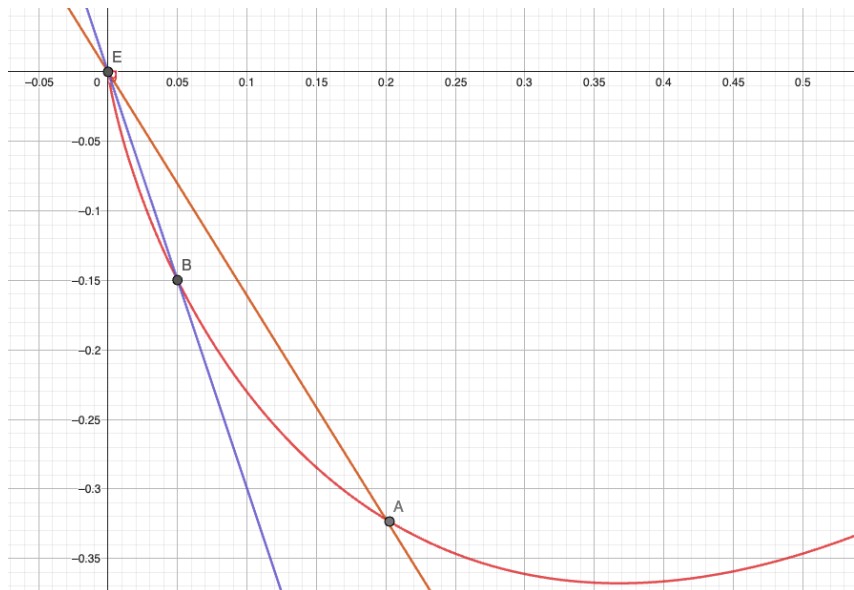

Figure 6: Illustration of the observation of Eq. 23 and Equ 24. The red, orange, and blue curves are $x \log x$, $\log \epsilon x$ and $\log \frac{\epsilon}{n-2} x$ (where $n = 6$ and $\epsilon = 0.2$), respectively. It is clear that $x \log x \leq \log \epsilon x$ between point $E$ and $A$; $x \log x \leq \log \frac{1}{n-2} \epsilon x$ between point $E$ and $B$.

By Assumption 5, we substitute terms appears in Eq. 19 with left hand side (LHS) of Eq. 12, further relax the bound as

$$\Delta > \lambda_1 \frac{\log \lambda_1 \epsilon}{\log (n-2)} - \frac{\log \epsilon}{\log (n-2)} + (1 - \lambda_1) + \frac{1 - \lambda_1}{\log n - 2} \log [(1 - \lambda_1)\epsilon] + \frac{1}{\log (n-2)}$$

$$= -\frac{1}{\epsilon \log (n-2)} [\lambda_1 \log \lambda_1 \epsilon + \epsilon \log \epsilon - \log (n-2)[(1 - \lambda_1)\epsilon] - (1 - \lambda_1)\epsilon \log (1 - \lambda_1)\epsilon - \epsilon]. \tag{20}$$

By multiplying $\epsilon \log (n-2)$ to both sides of the inequivality, we have

$$-\lambda_1 \epsilon \log (\lambda_1 \epsilon) + \epsilon \log \epsilon - [\log (n-2)](1 - \lambda_1 - \Delta)\epsilon - (1 - \lambda_1)\epsilon \log (1 - \lambda_1)\epsilon - \epsilon > 0. \tag{21}$$

Eq. 21 is equivalent to

$$-\lambda_1 \epsilon \log \lambda_1 \epsilon + \log \epsilon (\lambda_1 + \Delta)\epsilon + (n-2) \frac{(1 - \lambda_1 - \Delta)\epsilon}{n-2} \log \frac{\epsilon}{n-2} - (1 - \lambda_1)\epsilon \log (1 - \lambda_1)\epsilon - \epsilon > 0. \tag{22}$$

Now, we observe that since $\lambda_1 + \Delta = \lambda_2 < 1$, thus $(\lambda_1 + \Delta)\epsilon < \epsilon$. Therefore

$$(\lambda_1 + \Delta)\epsilon \log (\lambda_1 + \Delta)\epsilon \leq -\log \epsilon (\lambda_1 + \Delta)\epsilon. \tag{23}$$

Here, the $\log \epsilon$ is the slope of the linear function composed by $(0, 0)$ and $(\epsilon, \epsilon \log \epsilon)$. Analogously, we have

$$\frac{1 - \lambda_1 - \Delta}{\epsilon} \log \frac{1 - \lambda_1 - \Delta}{n-2} \epsilon \leq \log \frac{\epsilon}{n-2} \frac{(1 - \lambda_1 - \Delta)\epsilon}{n-2}. \tag{24}$$

By substituting the terms of RHS of Equ 23 and Equ 24 appeared in Equ 22 with the LHS of Equ 23 and Equ 24, we further relax our inequality as

$$-\lambda_1 \epsilon \log \lambda_1 \epsilon + (\lambda_1 + \Delta)\epsilon \log (\lambda_1 + \Delta)\epsilon + (1 - \lambda_1 - \Delta)\epsilon \log (\frac{1 - \lambda_1 - \Delta}{n-2} \epsilon) -$$
$$(1 - \lambda_1)\epsilon \log (1 - \lambda_1)\epsilon + (1 - \epsilon) \log 1 - \epsilon > 0. \tag{25}$$

We now rearrange the Eq. 25 and substitute $\lambda_1 + \Delta$ with $\lambda_2$, we have

$$-\lambda_1 \epsilon \log \lambda_1 \epsilon - (1 - \lambda_1)\epsilon \log (1 - \lambda_1)\epsilon >$$
$$-(\lambda_2 \epsilon) \log (\lambda_2 \epsilon) - (1 - \lambda_2)\epsilon \log (\frac{1 - \lambda_1 - \Delta}{n-2} \epsilon) - (1 - \epsilon) \log (1 - \epsilon). \tag{26}$$

We observe that, by Lemma 1, we have that

$$
\begin{aligned}
\min_{(x_1,\ldots,x_{n-2})} \Sigma_{\Sigma x_i = (1-\lambda_2)\epsilon} x_i \log x_i &= (1-\lambda_2)\epsilon \log\left(\frac{1-\lambda_2}{n-2}\epsilon\right), \\
\max_{(x_1,\ldots,x_{n-2})} \Sigma_{\Sigma x_i = (1-\lambda_1)\epsilon} x_i \log x_i &= (1-\lambda_1)\epsilon \log\left(1-\lambda_1\epsilon\right).
\end{aligned}
\tag{27}
$$

In other words,

$$
\begin{aligned}
\max_{(x_1,\ldots,x_{n-2})} |\Sigma_{\Sigma x_i = (1-\lambda_2)\epsilon} x_i \log x_i| &= -(1-\lambda_2)\epsilon \log\left(\frac{1-\lambda_2}{n-2}\epsilon\right), \\
\min_{(x_1,\ldots,x_{n-2})} |\Sigma_{\Sigma x_i = (1-\lambda_1)\epsilon} x_i \log x_i| &= -(1-\lambda_1)\epsilon \log\left(1-\lambda_1\epsilon\right).
\end{aligned}
\tag{28}
$$

Besides, it is direct to show that

$$
(1-\epsilon)\log\left(1-\epsilon\right) \le x_{\hat{a},i} \log x_{\hat{a},i} \le 0,
\tag{29}
$$

i.e.,

$$
-(1-\epsilon)\log\left(1-\epsilon\right) \ge |x_{\hat{a},i} \log x_{\hat{a},i}| \ge 0,
\tag{30}
$$

Hence, we rewrite the Eq. 26 to

$$
\begin{aligned}
&|\lambda_1\epsilon \log \lambda_1\epsilon| + \min_{(x_1,\ldots,x_{n-2})} |\Sigma_{\Sigma x_i = (1-\lambda_1)\epsilon} x_i \log x_i| + \min |x_{\hat{a},1} \log x_{\hat{a},1}| > \\
&|(\lambda_2\epsilon) \log\left(\lambda_2\epsilon\right)| + \max_{(x_1,\ldots,x_{n-2})} |\Sigma_{\Sigma x_i = (1-\lambda_2)\epsilon} x_i \log x_i| + \max |x_{\hat{a},2} \log x_{\hat{a},2}|.
\end{aligned}
\tag{31}
$$

Therefore, we are able to write that

$$
|\lambda_1\epsilon \log \lambda_1\epsilon + \Sigma_{\Sigma_i x_i = (1-\lambda_1)\epsilon} x_i \log x_i + x_{\hat{a},1}| \ge |\lambda_2\epsilon \log \lambda_2\epsilon + \Sigma_{\Sigma_i x_i = (1-\lambda_1)\epsilon} x_i \log x_i + x_{\hat{a},2}|,
\tag{32}
$$

which is exactly Eq. 17. $\qquad\square$

**Theorem 2.** Randomly flipping $K$ entries from a sequence of length $N$ will always keep the rank correlation within a range characterized by the lower bound $1 - 6\sum_{i=1}^{K}(a_i - a_{K+1-i})^2/N(N^2-1)$ and upper bound $1$. Here $a_i$ is the original position index of the $i$-th perturbed element. The lower bound is achieved with a probability of $1/K!$ when the perturbed sequence is the reverse of the original sequence. The upper bound is achieved with a probability of $1/K!$ when the perturbed sequence is identical to the original sequence.

To prove the above theorem, we first present the lemma:

**Lemma 2.** *Given a list of $N$ integers $\{a_1, a_2, \ldots, a_N\}$ with $a_i < a_{i+1}, i = 1, 2, \ldots, N-1$ and its random perturbation $\{a_1^*, a_2^*, \ldots, a_N^*\}$, the maximum value of $\sum_{i=1}^{N}(a_i - a_i^*)^2$ is achieved by reversing the list, i.e., $a_i^* = a_{N+1-i}$.*

*Proof.* To prove that the maximum value of the sum:

$$
S = \sum_{i=1}^{N}(a_i - a_i^*)^2
$$

is achieved by reversing the list $\{a_i^*\}_{i=1}^{N}$, we need to show that this arrangement maximizes the squared differences between the original list $\{a_i\}_{i=1}^{N}$ and the perturbed list $\{a_i^*\}_{i=1}^{N}$, where $a_i^*$ is the perturbed element in the $i$-th position.

We know that

$$
a_1 < a_2 < \cdots < a_N.
$$

Considering the sum $S = \sum_{i=1}^{N}(a_i - a_i^*)^2$, each term in this sum is of the form $(a_i - a_i^*)^2$, which measures how far apart $a_i$ and $a_i^*$ are. Thus, to maximize the sum, we need to maximize each individual squared difference $(a_i - a_i^*)^2$.

The largest possible difference between any two elements of the list $\{a_i\}_{i=1}^N$ occurs when the largest element $a_N$ is paired with the smallest element $a_1$, the second largest element $a_{N-1}$ is paired with the second smallest element $a_2$, and so on. In other words, the maximum possible difference occurs when $a_i^* = a_{N+1-i}$ for all $i$. This arrangement is precisely the reverse of the original list.

To prove that reversing the list maximizes the sum, we propose to prove that when swapping any two elements in the perturbed list, the sum will always decrease. Suppose we swap two elements $a_p^*$ and $a_q^*$ (with $p < q$, without loss of generality) in the reversed list. Before the swap, the contributions to the sum from the two positions are:

$$(a_p - a_p^*)^2 + (a_q - a_q^*)^2.$$

After swapping $a_p^*$ and $a_q^*$, the new contributions become:

$$(a_p - a_q^*)^2 + (a_q - a_p^*)^2.$$

The change in the sum, $\Delta S$, is the difference between these two expressions:

$$\Delta S = \left((a_p - a_q^*)^2 + (a_q - a_p^*)^2\right) - \left((a_p - a_p^*)^2 + (a_q - a_q^*)^2\right).$$

We expand these terms as follows:

- Before the swap:

$$(a_p - a_p^*)^2 + (a_q - a_q^*)^2 = (a_p - a_{N+1-p})^2 + (a_q - a_{N+1-q})^2$$

- After the swap:

$$(a_p - a_q^*)^2 + (a_q - a_p^*)^2 = (a_p - a_{N+1-q})^2 + (a_q - a_{N+1-p})^2$$

Because $a_p < a_q$ and the list is ordered, swapping two elements in the reversed list *decreases* the squared differences, leading to a decrease in the sum $S$. Thus, reversing the list maximizes the absolute differences $|a_i - a_i^*|$ for all $i$, and any deviation from the reversed order will result in a smaller sum. $\square$

With this lemma, now we prove Theorem 2.

*Proof.* Given two ranking sequences $\{s_i\}_{i=1}^N$ and $\{s_i^*\}_{i=1}^N$, the Spearman's rank correlation coefficient is represented as follows:

$$\rho = 1 - \frac{6\sum_{i=1}^N (s_i - s_i^*)^2}{N(N^2 - 1)}. \tag{33}$$

In our case, one ranking sequence is obtained by perturbing $K$ elements in another ranking sequence. Denote the selected elements as $\{a_i\}_{i=1}^K$, and the elements after perturbation as $\{a_i^*\}_{i=1}^K$ according to Lemma 2, we know the maximum value of $\sum_{i=1}^K (a_i - a_i^*)^2$ is achieved when $a_i^* = a_{K+1-i}$. For other elements that are not perturbed satisfy that their $d_i$ equals 0. Therefore, the Spearman's rank correlation coefficient reaches the minimum value:

$$\rho_{\min} = 1 - \frac{6\sum_{i=1}^K (a_i - a_{K+1-i})^2}{N(N^2 - 1)}. \tag{34}$$

Similarly, the maximum value is $\rho_{\max} = 1$ when the perturbed sequence is exactly the same as the original sequence. Since each perturbation has an equal probability, and there are $K!$ different perturbations, we know the probabilities are both $1/K!$. $\square$

# C SUPPLEMENTARY EXPERIMENTS

## C.1 ABLATION EXPERIMENT RESULTS

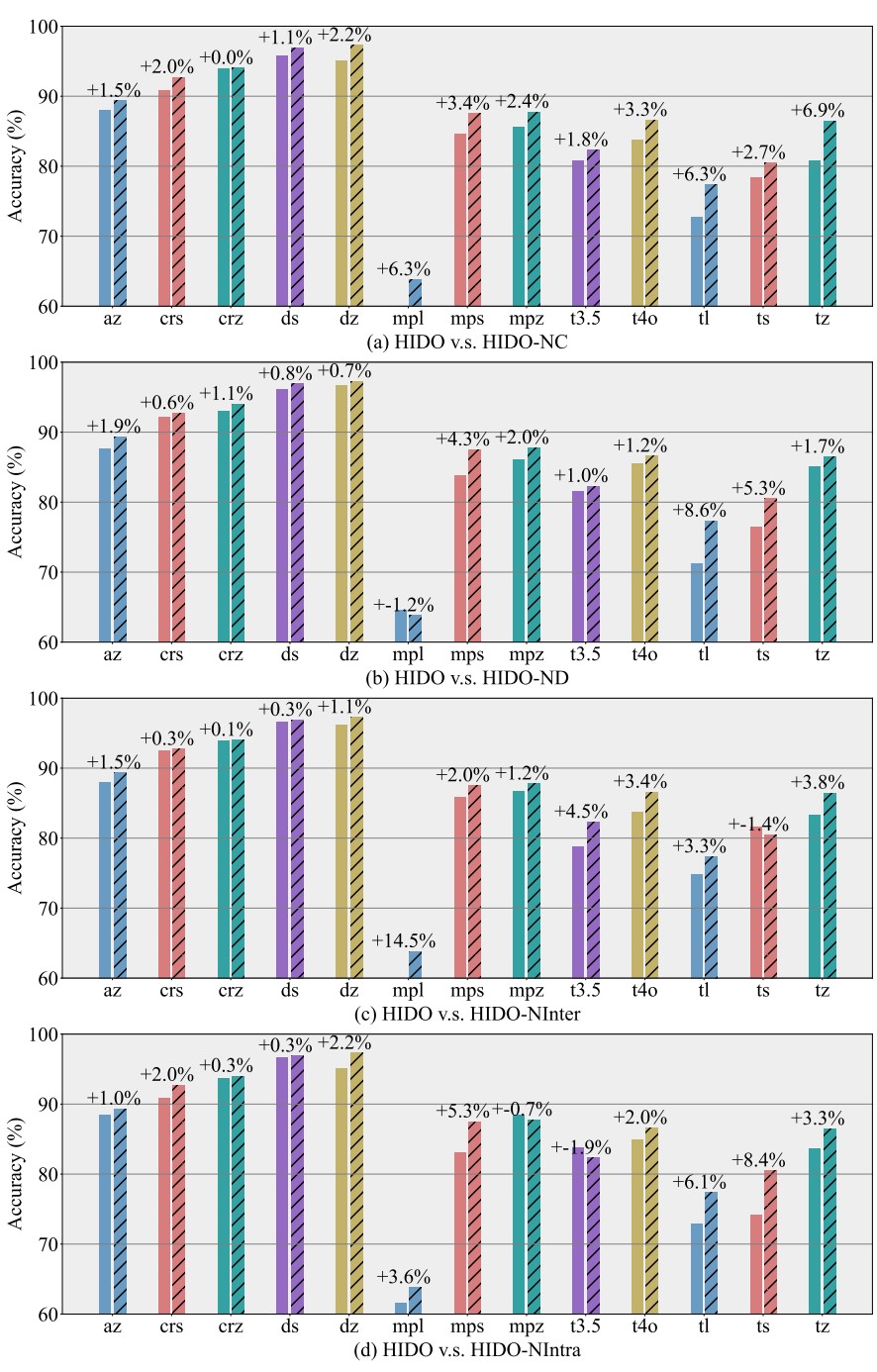

Figure 7: The performance of our proposed HIDO and its variants tested with different LLMs on various datasets. The first one or two characters indicate the dataset (i.e. 't' represents TREC and 'mp' represents 'MPQA'). The remaining characters represent the model (i.e. 'z' represents Zephyr and '3.5' represents GPT-3.5T).

## C.2  ACCURACY DIFFERENCE BETWEEN FEW-SHOT (10 SHOTS) AND MANY-SHOT (150) ICL

As mentioned earlier, we want to confirm that ICL-DOI still exists in many shot ICL. Thus, we randomly select orders with 10 or 150 demonstrations and measure the model accuracy. The following figures present the distribution of model performance under few-shot and many-shot settings on various datasets.

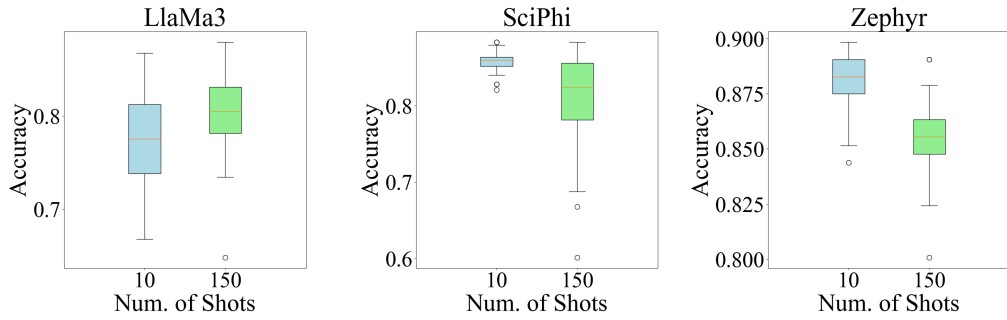

Figure 8: AGNews. Many shot ICL generally improves the best model accuracy (i.e. increases maximum accuracy), which causes the range to be larger.

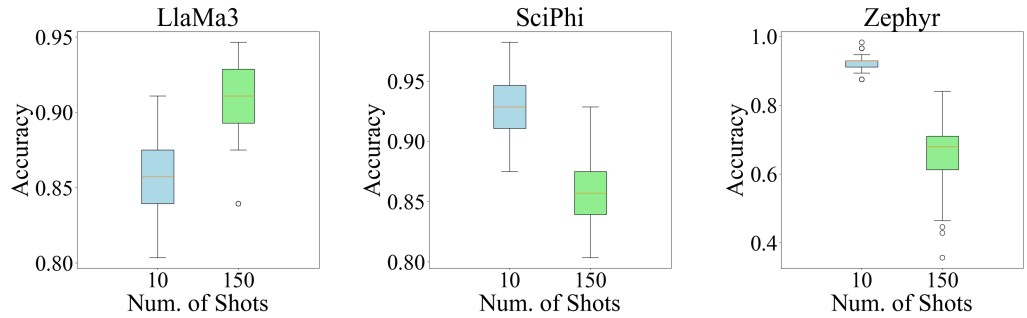

Figure 9: CB. Here, the figure shows that many shot learning causes model performance to degrade. This could be a result of CB having less test samples (56 samples compared to 256 samples for other datasets). Regardless, there is large variance in the results, indicating demonstration order instability.

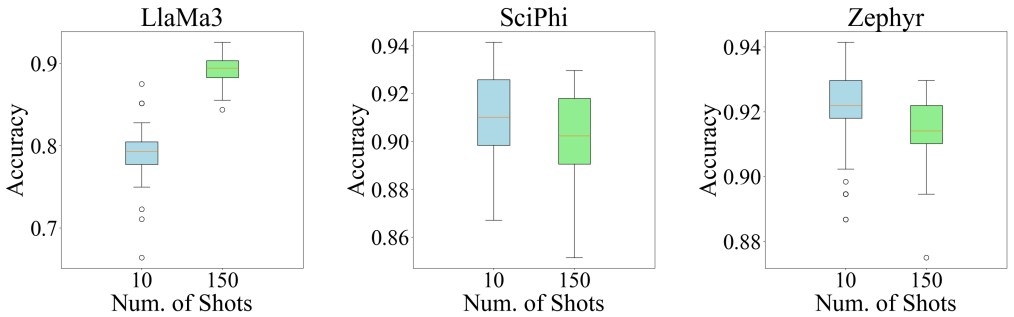

Figure 10: CR. SciPhi and Zephyr exhibit a wider variance in accuracy. In Zephyr, there is an extremely low outlier, emphasizing the importance of order on model performance.

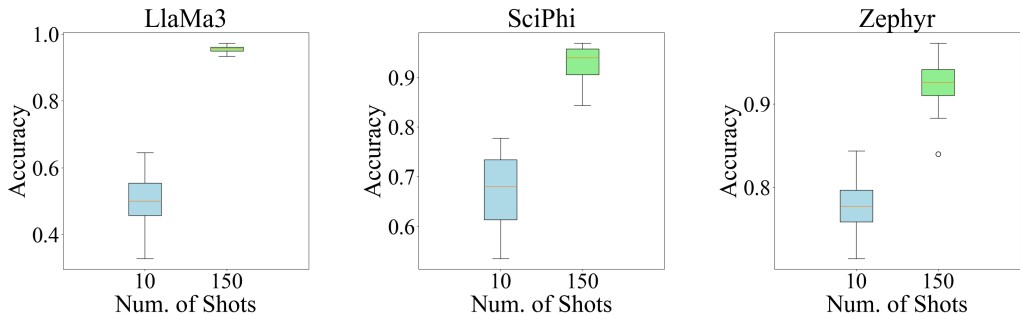

Figure 11: DBPedia. Many shot learning improved model performance for all models; however, for LlaMa3, the variance becomes smaller but stays the same or increases for the other models. Taking a look at DBPedia, the samples in general give more context in comparison to the others, which suggests that LlaMa3 is better at retaining and exploiting the information given from the demonstrations when completing the task of interest.

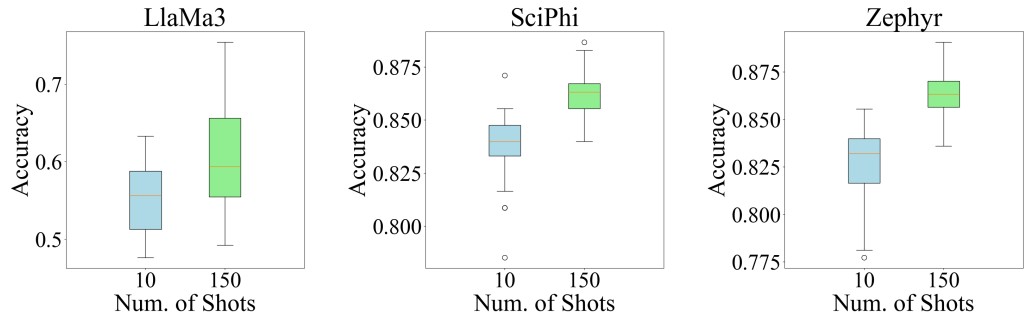

Figure 12: MPQA. Again, many shot ICL improved model accuracy, but also caused the variance to increase in general. LlaMa3 especially exhibits the problem of ICL-DOI with over 25% difference between the best and worst accuracy.

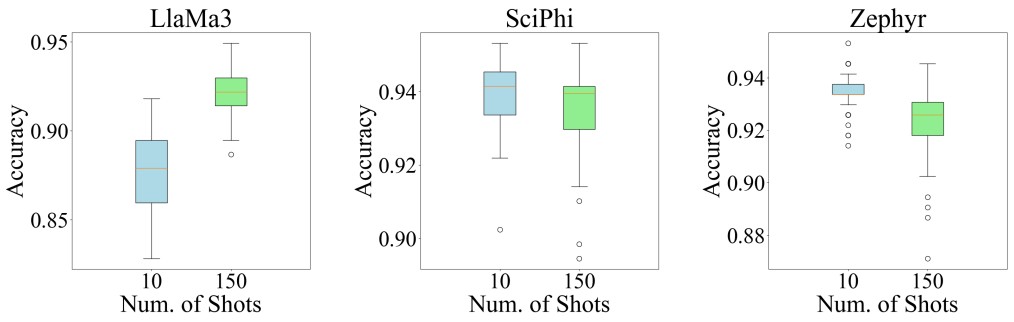

Figure 13: MR. Model performance only improved for LlaMa3, but the other two models illustrate a wider variance. For SciPhi and Zephyr, the model performance under the few-shot and many shot settings is comparable, but in many-shot, the worst accurcy is much lower than that of few-shot performance.

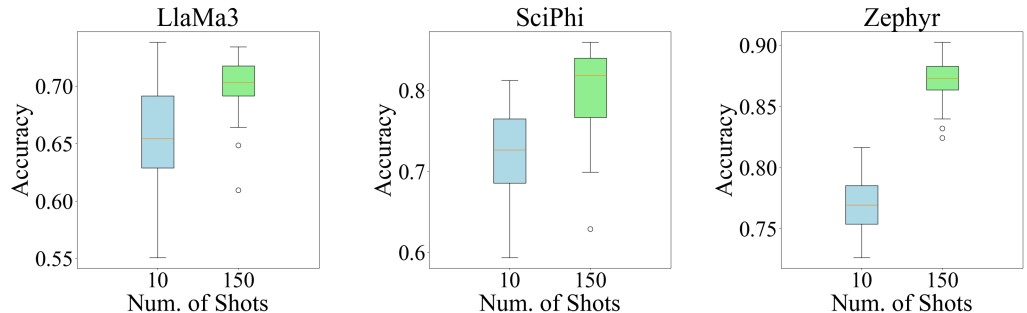

Figure 14: TREC. Increasing the number of demonstrations increased average accuracy for all models, and the variance did not improve much, other than for LlaMa3. LlaMa3 has 8 billion paramters, compared to only 7 billion for the other two models, which means that it has more capability to learn and retain information. This can potentially be the reason for its superior performance against the other two LLMs.

## C.3 QUANTITATIVE ANALYSIS OF GENERATED PROBING SETS

In Sec 4.1, we assume that the demonstration order optimized for answer prediction can also be used for sample generation. Since each additional iteration of HIDO optimizes the order such that it can achieve a higher accuracy, the probing set from the inter-cluster optimization round is generated from the current optimized order. Thus, we can compare the probing set to the original demonstrations, which should be of high quality. Ideally, as the number of iterations increases (i.e. the order becomes more optimized), the distance between the two should decrease (i.e. the quality of the probing set increases). The following figures measure the average $L_2$ norm between the demonstration embeddings and the probing set embeddings generated by various LLMs on different datasets. In general, the experiments support the assumption, presenting a negative trend between iterations and distance.

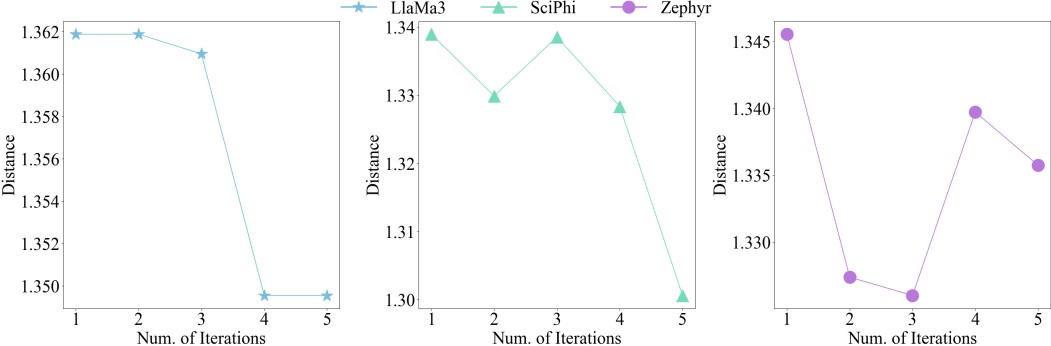

Figure 15: AGNews. Embedding distance for both LlaMa3 and Zephyr consistently decrease as the number of iterations increase; however, Zephyr reaches its optimal at three iterations, and additional iterations will cause the resulting order to deviate, as indicated by the spike at the fourth iteration. SciPhi has a peak at three iterations but decreases after that point.

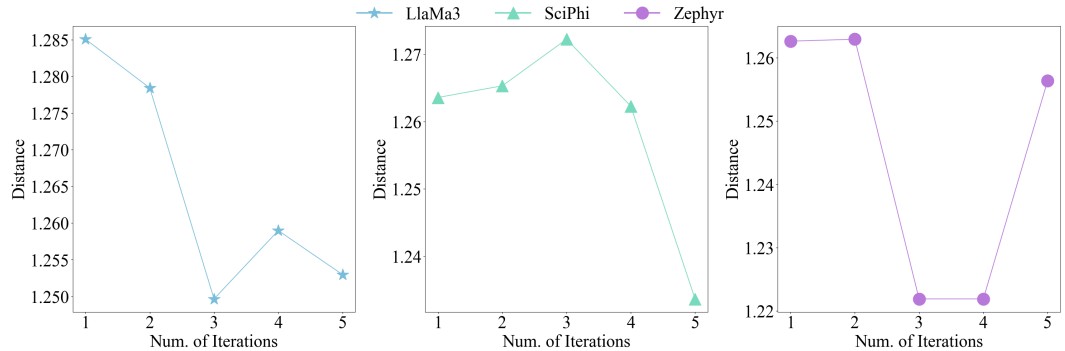

Figure 16: CR. Similar to the previous figure, the probing sets generated by LlaMa3 and Zephyr consistently drop, and SciPhi displays a peak and then a major drop in embedding distance. The figures suggest that after some iterations (i.e. as the order becomes more optimal), the LLM can generate samples close to the original text.

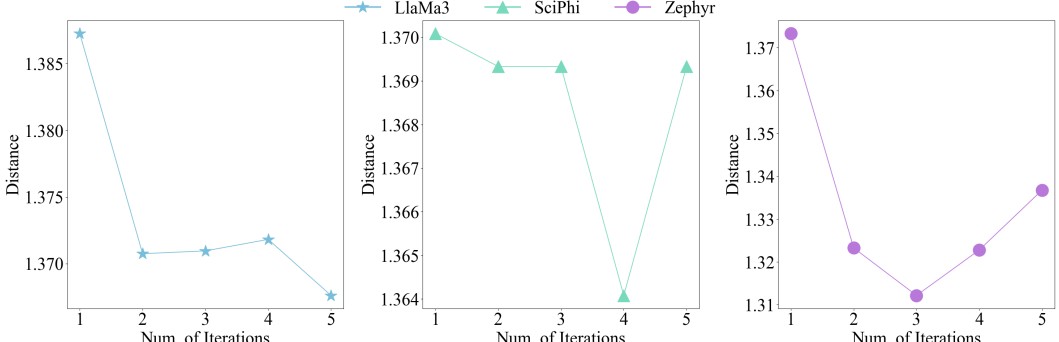

Figure 17: DBPedia. All models demonstrate a negative trend between distance and iteration. The figure for SciPhi displays a plateau between the second and third iteration, which could imply that the probing set (i.e. the actual text) or the semantics did not change much.

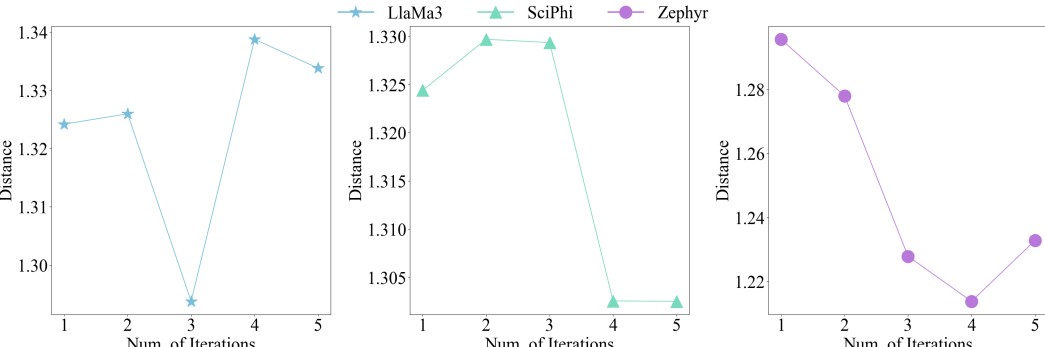

Figure 18: MPQA. The figures in general demonstrate a negative trend. For SciPhi, the distance increases first then drops after the second iteration. However, the difference is relatively small, about 0.05 difference, indicating that the generated samples are similar to the demonstrations.

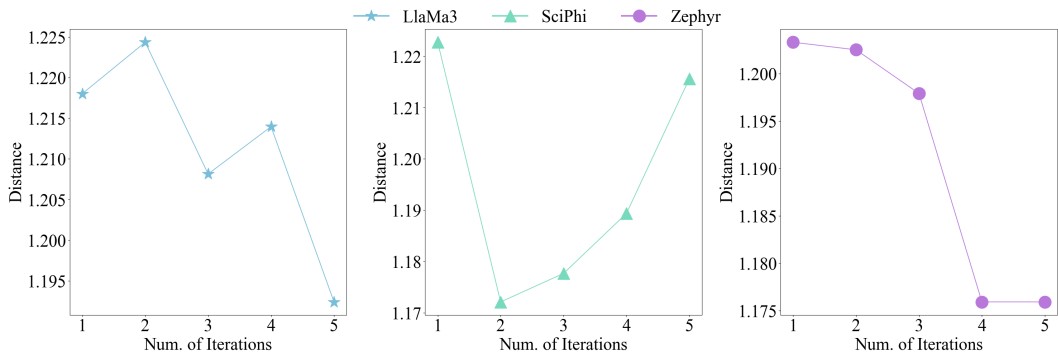

Figure 19: MR. For LlaMa3, the distance peaks at iteration two and iteration four, but generally decreases. This could be due to HIDO trying to find the best order in the neighborhood space but selecting one that does not perform well; however, it is able to find the best order in the end.

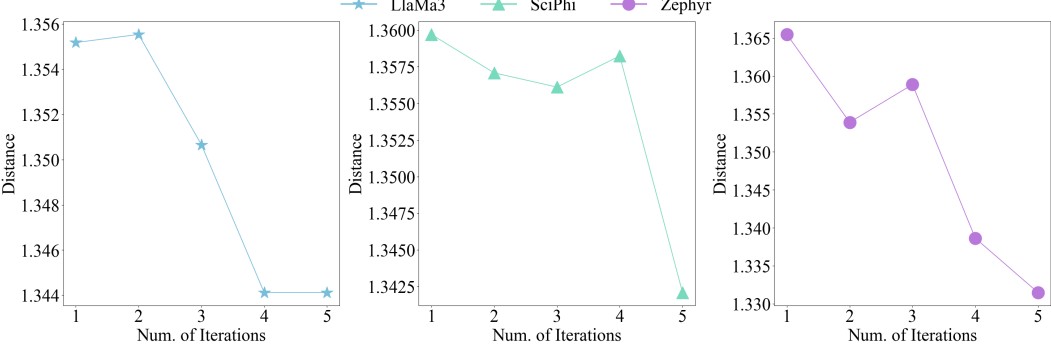

Figure 20: TREC. Like before, the genearl trend is negative in all the figures. However, the plots for SciPhi and Zephy both have a peak but drops in the next iteration, which indicates that the model diverges from the optimial and corrects itself.

## C.4 EXAMPLE SAMPLES FROM EACH DATASET

Below, we provide some samples in each dataset, which can be compared to the probing sets presented in C.5.

Table 3: Example Samples in AGNews

| Query | Label |
|---|---|
| AOL: Is Half a Billion Enough?. "At its height, the combined AOL and Time Warner (NYSE: TWX) company market capitalization exceeded $350 billion. When AOL used its big, bloated equity to buy Time Warner, the company offered Time Warner shareholders" | business |
| Justices to hear Seattle newspapers #39; dispute. Washington state #39;s highest court has agreed to review a key issue in a contentious lawsuit that could determine whether Seattle continues to have two daily newspapers. | business |
| Merck should have pulled Vioxx in 2000, study concludes. "Merck amp; Co. #39;s Vioxx painkiller showed heart risk in studies four years before the drug was recalled, and it should have been pulled from the market then, according to a study published in the medical journal Lancet." | business |
| Apple Extends iTunes to Europe. "The EU iTunes Music Store retains the same features and per-song price of 99 euro cents, established in June for customers in UK, Germany and France." | technology |
| Microsoft Introduces Fingerprint Recognition. "Microsoft rolled out an updated line of input devices, including its first fingerprint-recognition products. A mouse and keyboard with built-in fingerprint readers, along with a stand-alone reader " | technology |
| Justice Department Cracks Down On Spammers. It disrupted a network allegedly used to illegally share copyrighted files and is making a series of arrests against purveyors of spam. | technology |
| Red Sox Advance. David Ortiz homered in the 10th inning to give the Red Sox an 8-6 victory over Anaheim and completing a three-game sweep. | sports |
| Drugs in Sport: Baseball rocked by Bonds and Giambi admissions. "The ever-widening Balco scandal has sent new shockwaves through baseball, as it emerged that Barry Bonds, arguably the best player in the game, had used suspicious substances " | sports |
| Expect more discipline, warns Windies #39; CEO. "AS the West Indies players gather in Barbados for their three-week camp ahead of the short tour to Australia, an official of the West Indies Cricket Board (WICB) has warned the players to expect " | sports |
| Diplomatic push on N Korea talks. Intense diplomatic efforts are under way to persuade North Korea to give up its nuclear weapons programme. | world |
| Commonwealth chief meets Indian foreign minister (AFP). "AFP - Commonwealth Secretary General Don McKinnon held talks with Indian Foreign Minister Natwar Singh on the first day of his two-day visit to New Delhi, an Indian official said." | world |
| Ralph Klein leads Alberta Tories to 10th consecutive majority government (Canadian Press). "Canadian Press - EDMONTON (CP) - Alberta's Progressive Conservatives set a new benchmark for electoral success Monday, winning a record 10th consecutive majority government." | world |

Table 4: Example Samples in CB

| Query | Label |
|---|---|
| What must it be like to be imprisoned here, day after day, month after month? I wonder, does he keep them chained and manacled, thought Fenella, or does he use sorcery? And so utterly immersed was she in this strange blue and green land that was not feeling strange any more that she did not even notice that she was weighing sorcery against steel chains and seriously considering the likely outcome. | true |
| How did Selden know that the hound was following him? We know he ran a long way. He was screaming for a long time before he fell and we could hear that he was running as he screamed. | true |
| I didn't really like the way the other boys treated him. I was new at the school and still observing, still beginning friendships. Perhaps Alec noticed that I did not ridicule him as the others did. | true |
| A: Oh, wow! But maybe you shouldn't be held responsible for something you did several years ago. B: So, I know. A: That's the other thing. I mean a lot of people as kids or, you know, young people get into some things that they get out of later on and I don't think they should really have to pay for that forever. | false |
| B: What you want. where do they get it?. A: Well, I don't know, I guess they don't have it at home, B: I can't imagine it would stay fresh long enough to, | false |
| B: She says that when her husband died oh, that my uncle had said that he would never put her in a rest home. So it's kind of, uh, I don't know. I mean, I don't think my parents would but she is getting pretty bad like she has to have like a little toilet right by her bed and, it's, A: Uh-huh. B: and my mom has to take care of her pretty much so it gets, I don't know. it's a hard decision, but I don't think I would do it to my parents personally. | false |
| "I hope you are settling down and the cat is well." This was a lie. She did not hope the cat was well. | neutral |
| Then it cried. It was another girl. I was a little disappointed but I could only hope that Celia was still a bit hazy from the drugs. | neutral |
| B: Right. And I'm sure that would make a big difference, too. You know, you've got, A: Yeah. Well, what about a voluntary program? Do you think that would be a good idea? | neutral |

Table 5: Example Samples in CR

| Query | Label |
|---|---|
| after several years of torture in the hands of at&t customer service i am delighted to drop them , and look forward to august 2004 when i will convert our other 3 family-phones from at&t to t-mobile ! . | negative |
| the fact that the " 0 " key is the space key for text input is a bit confusing , as many phones use the " # " key instead . | negative |
| also , when i dial an " 800 " number like an airline , it will not allow me to use number keys to navigate the prerecorded menu . | negative |
| i downloaded and updated the firmware , and followed the easy setup instructions . n6600 supports midi , wav , arm , mp3 etc . | positive |
| and the phone has a very cool feature which allows you to send images via a normal pop3/smtp e-mail account . | positive |

Table 6: Example Samples in DBPedia

| Query | Label |
|---|---|
| Bonnie Burton (born July 12 1972) is a San Francisco-based author journalist co-median actress and show host. From 2003 to 2012 she worked as a Senior Editor for Star Wars.com and staff writer for Star Wars Insider magazine and was the Senior Editor of the Official Star Wars Blog. | artist |
| The Capitol Skyline Hotel is a hotel located near the United States Capitol in Capi-tol Hill Washington D.C. Designed by Morris Lapidus the hotel opened in the early 1960s and was once part of the Best Western chain. | building |
| Omorgus alternans is a beetle of the Family Trogidae. It occurs in Australia. | animal |
| Chrysotoxum triarcuatum is a species of hoverfly endemic to the Canary Islands. | animal |
| Molla Mahmud is a village in Abbas-e Sharqi Rural District Tekmeh Dash District Bostanabad County East Azerbaijan Province Iran. At the 2006 census its popula-tion was 43 in 8 families. | location |
| Shuji Kondo (Kondō Shūji) is a Japanese professional wrestler. Prior to becoming a pro wrestler he played rugby. | sports |
| Jerome J. Randle (born May 21 1987) is an American professional basketball player who currently plays for Trabzonspor of the Turkish Basketball League. | sports |
| HSwMS Wachtmeister (10/26) was a destroyer of the Royal Swedish Navy during World War I. | transport |
| The Sikorsky S-69 was an experimental compound co-axial helicopter developed as the demonstrator of the Advancing Blade Concept (ABC) under US Army and NASA funding. | transport |
| Corticon Technologies Inc. is a Business Rule Management System software com-pany that provides enterprise software products designed to automate decision man-agement through use of a patented rules engine that does not require coding. | company |
| Melvin Reginald Knight (born July 30 1944) was the Minister of Energy of Alberta and a Progressive Conservative member of the Legislative Assembly of Alberta. | political |
| Jannat Ki Talash is a Pakistani film. | film |
| Planta Medica is a peer-reviewed medical journal covering medicinal plants and bioactive natural products of plant origin. According to the Journal Citation Reports the journal has a 2012 impact factor of 2.348. | book |
| Phadion Design Classics is a three volume set of reference books on industrial de-sign since the 1600s. It lists 999 objects that the editorial team chose as design classics. Alan Fletcher was the art director for the project. | book |
| Bulbophyllum henrici is a species of orchid in the genus Bulbophyllum. | plant |
| Monte Porche is a mountain of Marche Italy. | nature |
| Lincoln Lake is located in Glacier National Park in the U. S. state of Montana. Lincoln Lake is .25 miles (0.40 km) downstream from Lake Ellen Wilson but sits more than 1300 feet (400 m) lower in elevation. A series of cascades including Beaver Chief Falls can be found between the two lakes. | nature |
| The Silence in My Heart is the sixth installment in The Emo Diaries series of compi-lation albums released July 24 2001 by Deep Elm Records. As with all installments in the series the label had an open submissions policy for bands to submit material for the compilation and as a result the music does not all fit within the emo style. | album |
| Seoul Women's University is a private women's university in Seoul South Korea. | school |
| The Lyme Academy College of Fine Arts is an art college in Old Lyme Connecticut | school |

Table 7: Example Samples in MPQA

| Query | Label |
|---|---|
| at turns passive and inflexible | negative |
| because | negative |
| no support for | negative |
| remains optimistic | positive |
| it must be regarded as an unambiguous message by the international community that the rule of democracy will be upheld in every part of the world | positive |
| most frantic lobbying | positive |

Table 8: Example Samples in MR

| Query | Label |
|---|---|
| . curiously , super troopers suffers because it doesn't have enough vices to merit its 103-minute length . | negative |
| the filmmaker ascends , literally , to the olympus of the art world , but he would have done well to end this flawed , dazzling series with the raising of something other than his own cremaster . | negative |
| both shrill and soporific , and because everything is repeated five or six times , it can seem tiresomely simpleminded . | negative |
| topics that could make a sailor blush - but lots of laughs . | positive |
| a penetrating , potent exploration of sanctimony , self-awareness , self-hatred and self-determination . | positive |
| a spunky , original take on a theme that will resonate with singles of many ages . | positive |

Table 9: Example Samples in RTE

| Query | Label |
|---|---|
| The fight originated when Gilson Ramos da Silva, 21, a.k.a. "Gilson Aritana," a member of the ADA ("Amigos do Bairro") gang led other members into the "morro da Mineira" (Miner Hill) to sell drugs, Ricardo Teixeira Dias, a local police official said. The region is controlled by rival gang Comando Vermelho (Red Command), which does not approve of other gangs selling drugs in the region. Comando Vermelho members started attacking the rival members of ADA to protect their turf. | false |
| The curious Belgian compromise over the weed has some logic, even for a country which says it wants to reduce drug use. Surveys show that as many as 40 percent of the country's 10 million population has experienced cannabis and with the Dutch border an hour away for most of the population, some liberalisation seems inevitable. | false |
| To the south of Castle Hill rises the higher Gellert Hill ( 771 feet ), a steep limestone escarpment overlooking the Danube, which provides a panoramic view of the whole city. | false |
| The first kibbutz, Deganya, near the Sea of Galilee, was founded in 1910. | true |
| Brazilian cardinal Dom Eusbio Oscar Scheid, Archbishop of Rio de Janeiro , harshly criticized Brazilian President Luiz Inácio Lula da Silva after arriving in Rome on Tuesday. | true |
| As his jubilant nation cheered, Yunus told reporters in the capital of Dhaka that he wants "to work to create some more new things in the world" and would use the award money to start a company to produce inexpensive yet nutritious food for poor people and set up an eye hospital to treat impoverished patients. | true |

Table 10: Example Samples in SST5

| Query | Label |
|---|---|
| ong 's promising debut is a warm and well-told tale of one recent chinese immigrant 's experiences in new york city . | great |
| a stirring , funny and finally transporting re-imagining of beauty and the beast and 1930s horror films | great |
| funny , somber , absurd , and , finally , achingly sad , bartleby is a fine , understated piece of filmmaking . | great |
| the movie ends with outtakes in which most of the characters forget their lines and just utter ' uhhh , ' which is better than most of the writing in the movie . | bad |
| viewers of barney 's crushingly self-indulgent spectacle will see nothing in it to match the ordeal of sitting through it . | bad |
| we 're left with a story that tries to grab us , only to keep letting go at all the wrong moments . | bad |
| wiser souls would have tactfully pretended not to see it and left it lying there | okay |
| paul bettany playing malcolm mcdowell ? | okay |
| he 's worked too hard on this movie . | okay |
| almost everyone growing up believes their family must look like " the addams family " to everyone looking in ... " my big fat greek wedding " comes from the heart ... | good |
| what 's surprising about this traditional thriller , moderately successful but not completely satisfying , is exactly how genteel and unsurprising the execution turns out to be . | good |
| it lets you brush up against the humanity of a psycho , without making him any less psycho . | good |
| the film is so busy making reference to other films and trying to be other films that it fails to have a heart , mind or humor of its own . | critical |
| but this new jangle of noise , mayhem and stupidity must be a serious contender for the title . | critical |
| there must be an audience that enjoys the friday series , but i would n't be interested in knowing any of them personally . | critical |

Table 11: Example Samples in Subj

| Query | Label |
|---|---|
| tsai may be ploughing the same furrow once too often . | bias |
| as it stands it's an opera movie for the buffs . | bias |
| to say analyze that is de niro's best film since meet the parents sums up the sad state of his recent career . | bias |
| paravasu is the elder son of the great sage raibhya ( mohan agashe ) . | objective |
| separated from his wife , louis will be given the opportunity to find out what is most important . . . | objective |
| through intimate conversations with top japanese artists , scholars and devotees from all cultures and walks of life , we reveal the multi-faceted appeal of the anime world . | objective |

Table 12: Example Samples in TREC

| Query | Label |
|---|---|
| How does lightning travel ? | description |
| How does a scientific calculator work ? | description |
| What is witch hazel ? | description |
| What operating system do IBM-compatible machines use ? | entity |
| What is a fear of home surroundings ? | entity |
| What joins white wine to put the spritz in a Spritzer ? | entity |
| Who created Maudie Frickett ? | human |
| Who was the architect of Central Park ? | human |
| Name of heroine in " Scruples " ? | human |
| How many different vegetation zones are there ? | number |
| How long does cocaine stay in your system ? | number |
| When was the Brandenburg Gate in Berlin built ? | number |
| Which continent has the most roses ? | location |
| What is the tallest building in Japan ? | location |
| What is the name of the largest city in Chile , South America ? | location |

C.5 QUALITATIVE ANALYSIS OF GENERATED PROBING SETS

In addition to Appendix C.3, we display the generated probing sets, along with example samples from each dataset, to qualitatively analyze the generated text. Because the embeddings may not completely capture the semantics and syntax of the text, we want to use human evaluation to determine if the quality of the probing set improves.

Table 13: Generated Probing Set using LlaMa3 on AGNews. The probing set from the beginning iterations are of the same sample with no diversity, indicating low quality. At the fourth iteration, the probind set consists of different samples, meaning that quality increased, but some of the samples are labeled incorrectly. In the last iteration, the new labels better fit the generated samples, further increasing the quality of the probing set.

| Iteration | Query | Label |
|---|---|---|
| 1 | The new iPhone 12 Pro Max is expected to have a larger battery life than its predecessor. | technology |
| 2 | The new iPhone 12 Pro Max is expected to have a larger battery life than its predecessor. | technology |
| | The new iPhone 12 Pro Max is expected to have a larger battery life than its predecessor. | technology |
| | The new iPhone 12 Pro Max is expected to have a larger battery life than its predecessor. | technology |
| | The new iPhone 12 Pro Max is expected to have a larger battery life than its predecessor. | technology |
| | The new iPhone 12 Pro Max is expected to have a larger battery life than its predecessor. | technology |
| 3 | The new iPhone 12 Pro Max is expected to have a larger battery capacity than its predecessor. | technology |
| | The new iPhone 12 Pro Max is expected to have a larger battery capacity than its predecessor. | technology |
| | The new iPhone 12 Pro Max is expected to have a larger battery capacity than its predecessor. | technology |
| | The new iPhone 12 Pro Max is expected to have a larger battery capacity than its predecessor. | technology |
| | The new iPhone 12 Pro Max is expected to have a larger battery capacity than its predecessor. | technology |
| 4 | The United States is set to launch a new initiative aimed at reducing carbon emissions from the transportation sector. | world |
| | The new initiative will focus on promoting the use of electric vehicles and public transportation. | world |
| | The initiative will also encourage companies to adopt more sustainable practices in their supply chains. | world |
| | The goal is to reduce carbon emissions by 50% by 2030. | world |
| | The initiative will be led by the Department of Transportation and will involve collaboration with various stakeholders. | world |
| 5 | The United States is set to launch a new initiative aimed at reducing carbon emissions from the transportation sector. | world |
| | The new initiative will focus on promoting the use of electric vehicles and public transportation. | business |
| | The initiative will also encourage companies to adopt more sustainable practices in their supply chains. | world |
| | The goal is to reduce carbon emissions by 50% by 2030. | business |
| | The initiative will be led by the Department of Transportation and will involve collaboration with various stakeholders. | world |

Table 14: Generated Probing Set using SciPhi on AGNews. For the first iteration, the model generates a variety of samples but the labels are the same with some not matching the corresponding sample. In the next iteration, SciPhi generates a better label for repeated samples. Additionally, the samples makes more sense, such as "Apple to Open First Retail Store in India" changed to "Apple to Launch New MacBook Pro Models." In the third iteration, the LLM only generates one sample, but it contains very specific names. For the fourth iteration, SciPhi generates more diverse samples with correct labels, indicating a very high quality probing set. Lastly, in the last iteration, the sample is very vague, which means the quality dropped.

| Iteration | Query | Label |
|---|---|---|
| 1 | Microsoft to Acquire LinkedIn for $26.2 Billion. | world |
| | NASA's Mars Rover Finds Evidence of Ancient Water. | world |
| | Apple to Open First Retail Store in India. | world |
| | China's Tencent to Acquire Majority Stake in Snapchat. | world |
| | SpaceX Successfully Launches Falcon Heavy Rocket. | world |
| 2 | Microsoft to Acquire LinkedIn for $26.2 Billion. | business |
| | Apple to Launch New MacBook Pro Models. | technology |
| | Tesla Model 3 Production Delayed Due to Production Issues. | technology |
| | Trump Administration Imposes Sanctions on Russian Companies. | world |
| | Manchester United Signs Cristiano Ronaldo in Record Deal. | sports |
| 3 | AMD mobile Athlon 64 targets Intel Pentium-M. | technology |
| 4 | AMD releases new Athlon 64 mobile processor. | technology |
| | Apple introduces new iPod nano with video playback capabilities. | technology |
| | US and UK forces launch joint operation in Afghanistan. | world |
| | Tesla Motors unveils new electric sports car. | technology |
| | Google announces new privacy settings for user accounts. | technology |
| | China and India sign agreement to reduce border tensions. | world |
| 5 | Apple to hold special event on October 26th. | world |

Table 15: Generated Probing Set using Zephyr on AGNews. In the first iteration, Zephyr generates diverse samples, but the syntax for some samples is invalid, like " Facebook to launch cryptocurrency Libra in 2020" and "Amazon to split second headquarters between New York City and Northern Virginia." In the next iteration, the samples are still very diverse, and all samples follow the same syntax of a subject followed by a verb. However, the sample "Commonwealth chief meets Indian foreign minister (AFP)" is vague in who the chief is. For the third iteration, the LLM replaces the sample mentioned earlier with a more specific one, and this probing set has the highest quaility. In the fourth iteration, the labels are less diverse than before, indicating the quality decreased. For the last iteration, the samples become less logical (i.e. "Biden administration announces \$1 billion for electric school buses" becomes "Biden administration extends eviction moratorium").

| Iteration | Query | Label |
|---|---|---|
| 1 | Google launches new Pixelbook Go laptop. | technology |
| | Manchester City beats Southampton 3-1 in Premier League. | sports |
| | Facebook to launch cryptocurrency Libra in 2020. | technology |
| | Trump imposes tariffs on \$200 billion of Chinese goods. | world |
| | NASA's Mars rover Curiosity finds signs of past life. | technology |
| | Amazon to split second headquarters between New York City and Northern Virginia. | business |
| 2 | IMF forecasts growth in Sub-Saharan economies. | business |
| | Apple extends iTunes to Europe. | technology |
| | Commonwealth chief meets Indian foreign minister (AFP). | world |
| | Bernie braves the Bank. | sports |
| | Chinese arrive in Haiti to help restore order. | world |
| | AMD mobile Athlon 64 targets Intel Pentium-M. | technology |
| | Kabila visits ex-rebel territory. | world |
| 3 | IMF forecasts growth in Sub-Saharan economies. | business |
| | Chinese arrive in Haiti to help restore order. | world |
| | Second round of Pakistan-India talks in Nov-Dec. | world |
| | Bernie braves the Bank. | sports |
| | Apple extends iTunes to Europe. | technology |
| | AMD mobile Athlon 64 targets Intel Pentium-M. | technology |
| 4 | Apple introduces new MacBook Air with M1 chip. | technology |
| | NASA's Perseverance rover lands on Mars. | technology |
| | Pfizer asks FDA to authorize COVID-19 vaccine for teens. | business |
| | SpaceX launches 60 Starlink satellites. | technology |
| | Biden administration announces \$1 billion for electric school buses. | business |
| | Facebook to allow some users to switch to alternative algorithm. | technology |
| 5 | Apple releases new MacBook Air with M1 chip. | technology |
| | Pfizer asks FDA to authorize COVID-19 vaccine for teenagers. | world |
| | SpaceX launches 60 Starlink satellites. | technology |
| | U.S. Jobless claims fall to 473,000. | business |
| | NASA's Perseverance rover lands on Mars. | world |
| | Biden administration extends eviction moratorium. | world |

Table 16: Generated Probing Set using LlaMa3 on CR. Samples from iteration one and iteration two are very similar, but those in iteration two have more diversity in syntax (i.e. samples in iteration one are descriptions in the format [subject] is [adjective]. For the third iteration, there's only positive labeled samples, but many of the samples are in first person, which follows the dataset. Further iterations continue to improve the samples.

| Iteration | Query | Label |
|---|---|---|
| 1 | The customer service at AT&T is terrible. | negative |
|  | The software installation process for Symantec products is frustrating. | negative |
|  | Nokia's recent loss of market share is well-deserved. | negative |
|  | Nokia is the best choice for cell phone games and ringtone customization. | positive |
|  | The Hitachi camera is a quality product, not like the cheap junk made in China. | positive |
|  | The T-Mobile customer service representatives are helpful. | positive |
| 2 | The customer service at AT&T is terrible. | negative |
|  | The software installation process is frustrating. | negative |
|  | Nokia has great cell phone games and customizable ring tones. | positive |
|  | Nokia's recent loss of market share is well-deserved. | negative |
|  | This product is of high quality and not comparable to cheap junk. | positive |
|  | The signal reception is outstanding. | positive |
|  | T-Mobile is trying hard and has good customer service representatives. | positive |
| 3 | The camera takes amazing pictures, even in low light. | positive |
|  | The battery life is impressive, I can go a whole day without needing to charge it. | positive |
|  | The phone is easy to use and navigate, even for someone who's not tech-savvy. | positive |
|  | The sound quality is top-notch, I can hear every detail in my music. | positive |
|  | The phone's design is sleek and modern, it looks great in my hand. | positive |
| 4 | The camera takes amazing pictures, even in low light. | positive |
|  | The battery life is impressive, I can go a whole day without needing to charge it. | positive |
|  | The user interface is intuitive and easy to navigate. | positive |
|  | The phone has a sleek design and is comfortable to hold. | positive |
|  | The speakerphone is loud and clear, making it perfect for group calls. | positive |
| 5 | The camera takes amazing pictures, even in low light. | positive |
|  | The battery life is impressive, I can go a whole day without needing to charge it. | positive |
|  | The user interface is intuitive and easy to navigate. | positive |
|  | The sound quality is top-notch, even when using the speakerphone. | positive |
|  | The phone is lightweight and fits comfortably in my pocket. | positive |
|  | The camera has a great zoom feature, allowing me to capture distant objects with ease. | positive |

Table 17: Generated Probing Set using SciPhi on CR. As the number of iterations increases, the samples become more clear in expressing the sentiment. For example, the sample from the first iteration "The customer service was terrible, I had a really hard time getting my issue re- solved" becomes "The customer service was terrible, I had a lot of issues with my order" in the third iteration, which is more direct in conveying discontent.

| Iteration | Query | Label |
|---|---|---|
| 1 | I've been using this product for a while now and it has been a game changer for me. | positive |
| | The customer service was terrible, I had a really hard time getting my issue re-solved. | negative |
| | I love the sleek design and the ease of use of this device. | positive |
| | The battery life is terrible, I have to charge it multiple times a day. | negative |
| 2 | I've been using this product for a while now and it's been a game changer for me. | positive |
| | The customer service was terrible, I had a really hard time getting my issue re-solved. | negative |
| | This phone has a great camera and the battery life is impressive. | positive |
| | I'm really disappointed with the performance of this laptop, it's been causing a lot of issues. | negative |
| 3 | I've been using this product for a while now and it's been a great experience. | positive |
| | The customer service was terrible, I had a lot of issues with my order. | negative |
| | I love the design of this product, it's sleek and modern. | positive |
| | The battery life on this phone is terrible, I have to charge it multiple times a day. | negative |
| 4 | I've been using this product for a while now and it's been a great experience. | positive |
| | The customer service was terrible, I had a lot of issues with my order. | negative |
| | This phone has a great camera and the battery life is impressive. | positive |
| | I'm really disappointed with the performance of this laptop, it's not up to par with other brands. | negative |
| 5 | I've been using this phone for a while now and it's been great. | positive |
| | The camera on this device is amazing, I've taken some fantastic photos. | positive |
| | I'm really happy with the performance of this laptop, it's been reliable and fast. | positive |
| | I've had some issues with the battery life on this device, it doesn't last as long as I'd like. | negative |

Table 18: Generated Probing Set using Zephyr on CR. The samples in the first iteration contains some phrases that are questionable, like "making navigation a breeze" and "rich and detailed audio." In further iterations, the LLM modifies the samples to be more clear. An interesting observation is that the third and fourth iteration generate the same probing sets with the only difference being the order, possibly due to the optimized order for those rounds being similar.

| Iteration | Query | Label |
|---|---|---|
| 1 | The battery life is impressive and lasts all day with moderate usage. | positive |
| | The touchscreen is responsive and accurate, making navigation a breeze. | positive |
| | The camera takes clear and vibrant photos, even in low light conditions. | positive |
| | The sound quality is exceptional, with rich and detailed audio. | positive |
| | The user interface is intuitive and easy to use, with a clean and modern design. | positive |
| 2 | The battery life is impressive and lasts all day with moderate usage. | positive |
| | The touchscreen is responsive and accurate, making navigation a breeze. | positive |
| | The camera takes clear and vibrant photos, even in low light conditions. | positive |
| | The sound quality is exceptional, with rich and detailed audio. | positive |
| | The design is sleek and modern, with a premium feel in the hand. | positive |
| 3 | the battery life is impressive. | positive |
| | the touchscreen is responsive. | positive |
| | the camera takes clear photos. | positive |
| | the software is user-friendly. | positive |
| | the sound quality is excellent. | positive |
| | the design is sleek and modern. | positive |
| | the connectivity options are plentiful. | positive |
| | the display is bright and clear. | positive |
| 4 | the battery life is impressive. | positive |
| | the touchscreen is responsive. | positive |
| | the camera takes clear photos. | positive |
| | the design is sleek and modern. | positive |
| | the sound quality is excellent. | positive |
| | the software is user-friendly. | positive |
| | the connectivity options are plentiful. | positive |
| | the display is bright and clear. | positive |
| 5 | The battery life is impressive and lasts all day. | positive |
| | The touchscreen is responsive and easy to use. | positive |
| | The camera takes clear and vibrant photos. | positive |
| | The design is sleek and modern. | positive |
| | The sound quality is excellent, both for calls and music playback. | positive |
| | The software is intuitive and user-friendly. | positive |

Table 19: Generated Probing Set using LlaMa3 on DBPedia. The probing sets become more detailed as the number of iterations increases, including more specific names of different locations and iconic landmarks. The labels also become more diverse, indicating an increase in quality.

| Iteration | Query | Label |
|---|---|---|
| 1 | The sun is shining brightly today. | company |
| | The cat is sleeping on the couch. | animal |
| | The bookshelf is overflowing with books. | company |
| | The baby is crying loudly. | company |
| | The dog is barking incessantly. | company |
| | The flowers are blooming beautifully. | nature |
| | The car is parked on the street. | location |
| | The cake is deliciously moist. | company |
| | The baby is laughing uncontrollably. | company |
| 2 | The sun is shining brightly today. | company |
| | The new employee is very enthusiastic. | company |
| | The company is expanding its operations. | company |
| | The concert was a huge success. | company |
| | The new policy is causing controversy. | company |
| | The book is a bestseller. | company |
| | The team won the championship. | sports |
| | The new restaurant has great reviews. | company |
| | The artist is very talented. | company |
| | The movie is a classic. | film |
| 3 | The University of California, Berkeley is a public research university located in Berkeley, California. | school |
| | The Great Wall of China is a series of fortifications built along the northern borders of China. | nature |
| | The Mona Lisa is a famous painting by Leonardo da Vinci. | company |
| | The Eiffel Tower is a famous landmark in Paris, France. | building |
| | The Grand Canyon is a natural wonder located in Arizona, USA. | nature |
| | The Louvre Museum is a famous art museum in Paris, France. | company |
| 4 | The University of California, Berkeley is a public research university located in Berkeley, California. | school |
| | The Great Wall of China is a series of fortifications built along the northern borders of China. | nature |
| | The Mona Lisa is a famous painting by Leonardo da Vinci. | company |
| | The Eiffel Tower is a famous landmark in Paris, France. | building |
| | The Grand Canyon is a natural wonder located in Arizona, USA. | nature |
| | The Taj Mahal is a famous monument in India. | building |
| 5 | The University of California, Berkeley is a public research university located in Berkeley, California. | school |
| | The Eiffel Tower is a wrought-iron lattice tower located on the Champ de Mars in Paris, France. | building |
| | The Great Barrier Reef is the world's largest coral reef system, located in the Coral Sea, off the coast of Australia. | nature |
| | The Mona Lisa is a portrait painted by Leonardo da Vinci in the early 16th century. | company |

Table 20: Generated Probing Set using SciPhi on DBPedia. The first iteration probing set has a variety of samples, but the sample "The 2018 Winter Olympics, officially known as the XXIII Olympic Winter Games, were held from 9 to 25 February 2018 in Pyeongchang, South Korea" is labeled incorectly. Future iterations repeat the same sample, which means low quality probing sets.

| Iteration | Query | Label |
|---|---|---|
| | The Great Gatsby is a 1925 novel written by American author F. Scott Fitzgerald. | book |
| 1 | The 2018 Winter Olympics, officially known as the XXIII Olympic Winter Games, were held from 9 to 25 February 2018 in Pyeongchang, South Korea. | company |
| | The Catcher in the Rye is a novel by J.D. Salinger, first published in 1951. | book |
| 2 | The Great Gatsby is a 1925 novel written by American author F. Scott Fitzgerald. | book |
| 3 | The Great Gatsby is a 1925 novel written by American author F. Scott Fitzgerald. | book |
| | The Great Gatsby is a 1925 novel written by American author F. Scott Fitzgerald. | book |
| 4 | The 2018 Winter Olympics were held in Pyeongchang County South Korea from February 9 to February 25 2018. | company |
| | The Cure is an English rock band formed in Crawley West Sussex in 1976. | company |
| 5 | The Great Gatsby is a 1925 novel written by American author F. Scott Fitzgerald. | book |

Table 21: Generated Probing Set using Zephyr on DBPedia. Throughout the iterations, the probing sets maintain high diversity and detailed samples. There is a slight increase in quality as indicated by the sample from the third iteration: "The Silence in My Heart is a compilation album in the Emo Diaries series released in 2001 by Deep Elm Records. It features a variety of artists and does not strictly adhere to the emo genre." It is modified in the next iteration to become clearer in the reason for why the album has a variety of artists.

| Iteration | Query | Label |
|---|---|---|
| 1 | The Eiffel Tower is a wrought-iron lattice tower located on the Champ de Mars in Paris, France. | building |
| | The Great Barrier Reef is the world's largest coral reef system composed of over 290 reefs and 900 islands stretching for 234 kilometers over an area of approximately 34,400 square kilometers. | nature |
| 2 | The Silence in My Heart is a compilation album in the Emo Diaries series released in 2001 by Deep Elm Records. It features a variety of artists and does not strictly adhere to the emo genre. | album |
| | The Capitol Skyline Hotel is a historic building located near the United States Capitol in Washington D.C. It was once part of the Best Western chain. | building |
| | The Lyme Academy College of Fine Arts is a school located in Old Lyme, Connecticut, dedicated to fine arts education. | school |
| 3 | The Silence in My Heart is a compilation album in the series The Emo Diaries, released in 2001 by Deep Elm Records. It features a variety of emo-inspired music, as the label allowed for open submissions. | album |
| | Meridix Creative Inc. is a technology services company founded in 2003 and incorporated in the state of Illinois originally to broadcast live local high school sports online. It now offers a broad range of technology services. | company |
| 4 | Dream.ZONE.Achieve is the third studio album by American rapper Smoke DZA. | album |
| | Meridix Creative Inc. Was founded in 2003 and was incorporated in the State of Illinois originally to broadcast live local high school sports online though the company now offers a broad range of technology services. | company |
| | Bulbophyllum henrici is a species of orchid. | plant |
| 5 | The Silence in My Heart is a compilation album in the emo genre released in 2001 by Deep Elm Records. | album |
| | The Sikorsky S-69 was a compound co-axial helicopter developed as the demonstrator of the Advancing Blade Concept under US Army and NASA funding. | transport |
| | The Wall Street Journal Europe is a daily English-language newspaper that covers global and regional business news for Europe, the Middle East, and Africa. | book |

Table 22: Generated Probing Set using LlaMa3 on MPQA. As the number of iterations increase, the generated probing set consists of more specific samples. For instance, from the second iteration, one sample is "The weather forecast is not looking good." In the fourth iteration, the sample becomes "The weather forecast is predicting a heatwave," which conveys the reason for negative sentiment. Furthermore, the samples from the beginning iterations mainly focus on business and government, but later on, the probing set includes samples of various topics.

| Iteration | Query | Label |
|---|---|---|
| 1 | The new policy is a step in the right direction, but it needs more support from the community. | positive |
| | The lack of funding for the project is a major setback. | negative |
| | The new employee is a great addition to the team. | positive |
| | The company's response to the crisis was inadequate. | negative |
| | The new product is a game-changer in the industry. | positive |
| | The lack of transparency in the company's financial reports is concerning. | negative |
| 2 | The new policy is a step in the right direction. | positive |
| | The lack of funding for the project is a major setback. | negative |
| | The company's decision to cut costs is a smart move. | positive |
| | The government's response to the crisis is inadequate. | negative |
| | The new employee is a great addition to the team. | positive |
| | The weather forecast is not looking good. | negative |
| | The new product launch was a huge success. | positive |
| 3 | The new policy is a step in the right direction. | positive |
| | The lack of support for the initiative is concerning. | negative |
| | The company's commitment to sustainability is admirable. | positive |
| | The government's crackdown on dissent is alarming. | negative |
| | The voices criticizing the decision are growing louder. | negative |
| | The policy of isolation and stifling creativity is outdated. | negative |
| | The allegations of corruption are damaging to the company's reputation. | negative |
| 4 | The new policy is confusing and unclear. | negative |
| | The company's customer service is exceptional. | positive |
| | The new employee is struggling to keep up with the workload. | negative |
| | The new restaurant has a great atmosphere. | positive |
| | The weather forecast is predicting a heatwave. | negative |
| | The new software is user-friendly and efficient. | positive |
| | The new employee is not a good fit for the team. | negative |
| 5 | The new policy is confusing and unclear. | negative |
| | The customer service representative was friendly and helpful. | positive |
| | The company's financial reports are difficult to understand. | negative |
| | The new employee is struggling to keep up with the workload. | negative |
| | The new restaurant has a great atmosphere and delicious food. | positive |
| | The weather forecast is predicting a chance of rain. | negative |
| | The new employee is not a good fit for the team. | negative |

Table 23: Generated Probing Set using SciPhi on MPQA. Tbe samples become more detailed as the number of iterations increase. Comparing the first iteration to the last iteration, the samples from the last one are much longer, providing more context and conveying the sentiment clearer.

| Iteration | Query | Label |
|---|---|---|
| 1 | The new product is a game-changer. | positive |
| | The company's decision was a disaster. | negative |
| | The politician's speech was inspiring. | positive |
| | The customer service was abysmal. | negative |
| | The teacher's dedication was commendable. | positive |
| | The movie's plot was predictable. | negative |
| | The chef's culinary skills were exceptional. | positive |
| 2 | The new product is a game-changer. | positive |
| | The customer service was unhelpful and unprofessional. | negative |
| | The teacher's passion for the subject was infectious. | positive |
| | The government's response to the crisis was inadequate. | negative |
| | The restaurant's ambiance was cozy and inviting. | positive |
| | The movie's plot was confusing and disjointed. | negative |
| 3 | The new product is a game-changer. | positive |
| | The customer service was unhelpful and unprofessional. | negative |
| | The teacher's passion for the subject was contagious. | positive |
| | The government's response to the crisis was inadequate. | negative |
| | The restaurant's ambiance and service were exceptional. | positive |
| | The movie's plot was predictable and clich. | negative |
| 4 | The new policy is a step in the right direction. | positive |
| | The government's actions are causing chaos and confusion. | negative |
| | The company's decision to cut corners is unethical. | negative |
| | The president's speech was inspiring and hopeful. | positive |
| | The protestors' demands are justified and should be addressed. | positive |
| | The government's response to the crisis is inadequate. | negative |
| 5 | The new product is a significant improvement over its predecessor. | positive |
| | The company's decision to discontinue the product was met with widespread protest. | negative |
| | The government's policy of isolation and stifling has led to a decline in innovation. | negative |
| | The president's commitment to democracy is commendable. | positive |
| | The anti-terrorism campaign has been marred by allegations of human rights abuses. | negative |

Table 24: Generated Probing Set using Zephyr on MPQA. The generated labels first start out as all positive and, at the third iteration, the label distribution becomes more balanced. The samples gradually become shorter, which follows the format of the original dataset. This indicates that the LLM is able to mimic the syntax fromt he demonstrations and generate similar samples.

| Iteration | Query | Label |
|---|---|---|
| 1 | the product is a game-changer | positive |
| | the service exceeded expectations | positive |
| | the restaurant has a cozy atmosphere | positive |
| | the hotel is centrally located | positive |
| | the software is user-friendly | positive |
| | the book is a page-turner | positive |
| | the movie left me on the edge of my seat | positive |
| | the concert was a musical masterpiece | positive |
| 2 | delicious food and excellent service | positive |
| | highly recommended | positive |
| | clean and comfortable rooms | positive |
| | perfect location | positive |
| | great value for money | positive |
| | friendly and helpful staff | positive |
| | beautiful views | positive |
| | fantastic amenities | positive |
| | wonderful experience | positive |
| | highly rated by guests | positive |
| 3 | highly effective | positive |
| | flawed strategy | negative |
| | clear benefits | positive |
| | counterproductive actions | negative |
| | steadfast ally | positive |
| | peaceful resolution | positive |
| | hostile rhetoric | negative |
| | constructive dialogue | positive |
| | destructive policies | negative |
| | unwavering support | positive |
| 4 | harsh criticism | negative |
| | condemned | negative |
| | orchestrated | negative |
| | protested | negative |
| | most fervent lobbying | positive |
| | end of the era of violence and counterviolence | positive |
| | committed | positive |
| | staunch supporter | positive |
| | it is certainly playing games | negative |
| 5 | outdated | negative |
| | flawed | negative |
| | misguided | negative |
| | ill-conceived | negative |
| | counterproductive | negative |
| | detrimental | negative |
| | damaging | negative |
| | destructive | negative |
| | detrimental | negative |
| | misguided | negative |
| | erroneous | negative |

Table 25: Generated Probing Set using LlaMa3 on MR. In the earlier iterations, the generatd labels are incorrect. For example, "The plot twists and turns in this movie are so unpredictable that you'll be on the edge of your seat the whole time" is labeled negative when it conveys a positive sentiment. In the following iterations, the model generates correct labels for the corresponding sample except for the last iteration. This could indicate that the optimal number of iterations is three or four, and exceeding that value will cause the model to generate worse probing sets.

| Iteration | Query | Label |
|---|---|---|
| 1 | The plot twists and turns in this movie are so unpredictable that you'll be on the edge of your seat the whole time. | negative |
| | The special effects in this movie are so impressive that you'll forget you're watching a sci-fi film. | negative |
| | The acting in this movie is so bad that it's actually kind of entertaining. | negative |
| | The cinematography in this movie is so breathtaking that you'll forget you're watching a drama. | negative |
| 2 | The plot is predictable and lacks originality. | negative |
| | The acting is subpar and the dialogue is cringe-worthy. | negative |
| | The cinematography is stunning and the soundtrack is perfect. | negative |
| | The pacing is slow and the story is dull. | negative |
| | The characters are well-developed and the themes are thought-provoking. | negative |
| | The special effects are impressive but the plot is lacking. | negative |
| 3 | The plot twists in this movie are so confusing that it's hard to keep up. | negative |
| | The special effects in this movie are mind-blowing, but the storyline is lacking. | negative |
| | This movie is a waste of time, it's not worth watching. | negative |
| | The acting in this movie is superb, but the plot is predictable. | negative |
| | The movie's themes are thought-provoking, but the execution falls short. | negative |
| 4 | The plot is predictable and lacks originality. | negative |
| | The acting is subpar and the dialogue is cringe-worthy. | negative |
| | The special effects are impressive but the storyline is weak. | negative |
| | The movie is visually stunning but emotionally empty. | negative |
| | The pacing is slow and the characters are underdeveloped. | negative |
| | The humor is forced and the jokes fall flat. | negative |
| 5 | The plot twists in Blood Work are gripping, but the film's length feels excessive. | negative |
| | The performances in Dirty Deeds are captivating, but the premise is overshadowed by the violence. | negative |
| | The sword fighting in The Man in the Iron Mask is well done, but the film is ultimately a comedy. | negative |
| | Secret Ballot is a funny and puzzling movie that is both engaging and moving. | negative |

Table 26: Generated Probing Set using SciPhi on MR. The samples in the probing set for the first iteration is very direct, following the format of the subject followed by adjectives. As the number of iterations increase, the samples typically suggest a reason of the opposite sentiment before stating the main clause, such as "The film's visuals are stunning, but the plot is lackluster." This increases the complexity of the samples as it is harder to determine the sentiment.

| Iteration | Query | Label |
|---|---|---|
| 1 | The Martian is a thrilling and engaging sci-fi adventure. | positive |
| | The Hunger Games: Mockingjay Part 1 is a disappointing and slow-paced installment in the series. | negative |
| | The Grand Budapest Hotel is a visually stunning and witty comedy-drama. | positive |
| | The Lobster is a quirky and thought-provoking film. | positive |
| 2 | The film's exploration of sanctimony, self-awareness, self-hatred, and self-determination is both penetrating and potent. | positive |
| | The gantzes' interviews in the movie tend to let the guys off the hook, which is disappointing. | negative |
| | Godard's ideas about creation and identity in the film are not always profound, but he is still smarter than many other filmmakers. | positive |
| 3 | The film's exploration of sanctimony, self-awareness, self-hatred, and self-determination is both penetrating and potent. | positive |
| | The comedy in Van Wilder is overshadowed by a disgusting scene that may cause viewers to lose their lunch. | negative |
| | Godard's ideas about creation and identity in Weekend may not be profound, but his insights are still smarter than many other filmmakers. | positive |
| 4 | The film's visuals are stunning, but the plot is lackluster. | negative |
| | The acting is superb, but the storyline is predictable. | negative |
| | The movie is a thrilling ride, but the character development is lacking. | negative |
| | The cinematography is breathtaking, but the pacing is slow. | negative |
| | The film is a heartwarming tale, but the dialogue is clich. | negative |
| 5 | The film's visuals are stunning, but the plot is lackluster. | negative |
| | A heartwarming story that will leave you with a smile on your face. | positive |
| | The movie's pacing is slow and the characters are underdeveloped. | negative |
| | A thrilling adventure that keeps you on the edge of your seat. | positive |
| | The film's message gets lost in its convoluted plot. | negative |

Table 27: Generated Probing Set using Zephyr on MR. The syntax of the samples shift from complete sentences to incomplete sentences as the number of iterations increase. This is likely due to the original dataset being human reviews of movies, so the datset contains samples of this format. Note that the probing sets of the fourth and the fifth iteration ar ethe same, indicating that the LLM reached the optimal point in generating samples (or order optimization) in an earlier iteration.

| Iteration | Query | Label |
|---|---|---|
| 1 | The art direction in this film is stunning, transporting the viewer to a different time and place. | positive |
| | The pacing in this movie is slow and tedious, making it a real chore to sit through. | negative |
| | The lead actor delivers a captivating performance that will leave you on the edge of your seat. | positive |
| | The plot of this movie is predictable and lacks any real surprises. | negative |
| 2 | The art direction in this film is stunning, transporting the viewer to a different time and place. | positive |
| | The pacing in this movie is slow and tedious, making it a chore to sit through. | negative |
| | The lead actor delivers a captivating performance that will leave you on the edge of your seat. | positive |
| | The plot twists in this movie are predictable and unoriginal, making it a disappointment. | negative |
| 3 | a heartwarming and uplifting story that will leave you feeling inspired. | positive |
| | a visually stunning and emotionally powerful film that will stay with you long after the credits roll. | positive |
| | a hilarious and raunchy comedy that will have you laughing out loud. | positive |
| | a gripping and suspenseful thriller that will keep you on the edge of your seat. | positive |
| 4 | a heartwarming and uplifting story that will leave you feeling inspired. | positive |
| | a visually stunning and emotionally powerful film that will stay with you long after the credits roll. | positive |
| | a thrilling and suspenseful ride that will keep you on the edge of your seat. | positive |
| | a poignant and thought-provoking film that explores complex themes with depth and nuance. | positive |
| 5 | a heartwarming and uplifting story that will leave you feeling inspired. | positive |
| | a visually stunning and emotionally powerful film that will stay with you long after the credits roll. | positive |
| | a thrilling and suspenseful ride that will keep you on the edge of your seat. | positive |
| | a poignant and thought-provoking film that explores complex themes with depth and nuance. | positive |

Table 28: Generated Probing Set using LlaMa3 on TREC. The probing set from the first iteration have many repetitive questions. In the next iteration, the LLM generates a better set, removing the repeated samples. However, the questions are still similar to each other, asking for the largest entity. For the third iteration, the same problem persists. Lastly, the model only generates one sample for the fourth and fifth iteration, possibly because it cannot generate better samples without repeating itself.

| Iteration | Query | Label |
|---|---|---|
| | What is the name of the famous painting that depicts a woman with a mysterious smile? | entity |
| | Who is the lead singer of the band Coldplay? | human |
| 1 | What is the capital city of Australia? | location |
| | What is the highest mountain peak in the world? | entity |
| | What is the name of the famous painting that depicts a woman with a mysterious smile? | entity |
| | Who is the lead singer of the band Coldplay? | human |
| | What is the capital city of Australia? | location |
| | What is the name of the largest city in Brazil? | location |
| | What is the highest mountain in the world? | description |
| | Who is the lead singer of the band Coldplay? | human |
| 2 | What is the capital of Australia? | location |
| | What is the name of the largest planet in our solar system? | description |
| | What is the name of the largest desert in the world? | location |
| | What is the name of the largest river in South America? | location |
| | What is the name of the largest city in Brazil? | location |
| | Who is the lead singer of the band Coldplay? | human |
| | What is the highest mountain peak in the world? | location |
| 3 | What is the name of the famous painting by Leonardo da Vinci? | entity |
| | What is the name of the famous painting by Vincent van Gogh? | entity |
| | What is the name of the famous painting by Pablo Picasso? | entity |
| | What is the name of the famous painting by Claude Monet? | entity |
| 4 | What is the name of the famous painting that depicts a woman with a mysterious smile? | entity |
| 5 | What is the name of the famous painting that depicts a woman with a mysterious smile? | entity |

Table 29: Generated Probing Set using SciPhi on TREC. The topics of the probing sets remain relatively the same in the first few iterations. There is repetition within the probing sets, such as multiple questions about the capital of a state. In the last iteration however, there is muore variety in the samples, diverging from the topics of capitals and chemical symbols.

| Iteration | Query | Label |
|---|---|---|
| 1 | What is the capital of Australia? | location |
| | Who is the author of the novel "To Kill a Mockingbird"? | human |
| | What is the chemical symbol for gold? | entity |
| | What is the capital of Canada? | location |
| | Who is the lead singer of the band "Nirvana"? | human |
| | What is the scientific name for a shark? | entity |
| | What is the capital of France? | location |
| 2 | What is the capital of Australia? | location |
| | Who is the author of the novel "To Kill a Mockingbird"? | human |
| | What is the chemical symbol for gold? | entity |
| | Who was the first man to walk on the moon? | human |
| | What is the capital of France? | location |
| | Who is the lead singer of the band Nirvana? | human |
| | What is the largest organ in the human body? | entity |
| 3 | What is the capital city of Australia? | location |
| | Who is the author of the novel "To Kill a Mockingbird"? | human |
| | What is the chemical symbol for gold? | entity |
| | What is the capital city of France? | location |
| | Who is the lead singer of the band "Queen"? | human |
| | What is the chemical symbol for silver? | entity |
| | What is the capital city of Germany? | location |
| 4 | What is the capital city of Australia? | location |
| | Who is the author of the novel "To Kill a Mockingbird"? | human |
| | What is the chemical symbol for gold? | entity |
| | What is the capital city of Canada? | location |
| | Who is the lead singer of the band "The Rolling Stones"? | human |
| | What is the largest organ in the human body? | entity |
| | What is the capital city of France? | location |
| 5 | What constellation is known as The Water Bearer? | location |
| | What is witch hazel? | description |
| | How long does cocaine stay in your system? | number |
| | How much in miles is a ten K run? | number |
| | How many different vegetation zones are there? | number |
| | How many colors are there in a rainbow? | number |
| | How many stars are there on the Soviet Union's flag? | number |

Table 30: Generated Probing Set using Zephyr on TREC. When comparing different probing sets, the topics of the samples become more diverse with each additional iteration. Additionally, the samples increase in difficulty as the questions from the first iteration's probing set revolve around pop culture, but subsequent probing sets are more technical and require specialized knowledge.

| Iteration | Query | Label |
|---|---|---|
| 1 | What is the capital of Australia? | location |
| | Who played the role of Vivian Ward in the movie "Pretty Woman"? | human |
| | What is the name of the character played by Tom Hanks in the movie "Forrest Gump"? | human |
| | What is the name of the actress who portrayed Elizabeth Bennet in the 2005 adaptation of "Pride and Prejudice"? | human |
| | Who played the role of Katniss Everdeen in the Hunger Games movie series? | human |
| 2 | Who holds the record for most goals in a single soccer match? | human |
| | What is the name of the first man to walk on the moon? | human |
| | What is the capital city of Australia? | location |
| | What is the largest animal in the world? | entity |
| | What is the highest mountain in North America? | location |
| | Who painted the Mona Lisa? | human |
| 3 | Who is known as the "Father of Modern Medicine"? | human |
| | What is the scientific name for a virus that causes the common cold? | entity |
| | What is the name of the largest planet in our solar system? | location |
| | What is the chemical symbol for gold? | entity |
| | What is the name of the first man to walk on the moon? | human |
| | What is the name of the largest city in South America? | location |
| 4 | What is the fear of being in one's own home called? | description |
| | What novel by Peter Blatty tells the story of Regan MacNeil's demonic possession? | entity |
| | Who created Maudie Frickett? | human |
| | Who is known for shoplifting? | human |
| | What is the name of the main character in "Scruples"? | human |
| 5 | Who created the character Maudie Frickett? | human |
| | What is the name of the lead singer of the band Led Zeppelin? | human |
| | What is the highest-ranking suit in bridge? | entity |
| | What did Louis Cartier invent for aviator Santos Dumont in 1940? | entity |
| | What is a multiplexer? | description |
| | What song put James Taylor in the limelight? | entity |

