# OpenReview forum: "Hierarchical Demonstration Order Optimization for Many-shot In-Context Learning"
_ICLR.cc/2025/Conference — ICLR 2025 Conference Withdrawn Submission_

### Official Review · Reviewer_J7Po · 2024-10-31

**Soundness:** 4
**Presentation:** 3
**Contribution:** 3
**Rating:** 8
**Confidence:** 2

**Summary:**

The paper introduces HIDO, a hierarchical optimization method, and ICD-OVI, an information-theoretic metric, to address demonstration order instability (ICL-DOI) in large language models. HIDO reduces computational complexity by separately optimizing order within clusters and between clusters. Meanwhile, ICD-OVI measures the information gain from each order to help select the optimal sequence. Experimental results show that HIDO, with dynamic updates to ICD-OVI, outperforms baselines in both predictive accuracy and stability.

**Strengths:**

1. Innovative Method: The paper introduces HIDO, a novel hierarchical optimization approach that effectively addresses demonstration order instability (ICL-DOI) in many-shot in-context learning, meanwhile reducing the computational complexity of order optimization, a persistent challenge in large language models.
2. Theoretical Rigor: Introducing ICD-OVI, an information-theoretic metric, provides a robust, quantifiable measure of information gain for different demonstration orders, adding a solid theoretical foundation to the method.
3. Strong Experimental Validation: Extensive experiments on multiple datasets and models demonstrate that HIDO outperforms baseline methods in both predictive accuracy and stability, highlighting its practical effectiveness.

**Weaknesses:**

Although the method shows strong results on several datasets, additional validation on a broader range of tasks, especially outside of text classification, would strengthen the claim of its general applicability in many-shot in-context learning.

**Questions:**

Could you please specify the hardware used (e.g., type of processors/GPUs, memory) and the approximate computation time required to run the proposed HIDO method on the datasets in the experiments? Thank You

---

> ### Author Response · Authors · 2024-12-01
> **Author Response 1/1**
>
> We sincerely appreciate the time and efforts you've dedicated to reviewing and providing invaluable feedback to enhance the quality of this paper. We provide a point-to-point reply below for the mentioned concerns and questions.
> ## Weaknesses
> > 1. (Experiment) Additional validations out of text classifications are needed.
>
> |        |               |     HellaSwag    |       MMLU       |
> |:------:|:-------------:|:----------------:|:----------------:|
> | SciPhi |    GlobalE    |  _53.65 ± 5.67_  |   55.60 ± 0.23   |
> |        |     IGDemo    |   51.17 ± 0.00   |   53.91 ± 0.00   |
> |        |     LocalE    |   48.31 ± 1.19   |  _56.38 ± 1.85_  |
> |        |      PDO      |  _53.65 ± 5.67_  |   55.60 ± 0.23   |
> |        | RetrievalDemo |   52.73 ± 0.00   |   55.86 ± 0.00   |
> |        |      HIDO     | **61.07 ± 2.98** | **56.64 ± 1.35** |
> | Zephyr |    GlobalE    |   28.26 ± 3.69   |   43.10 ± 0.81   |
> |        |     IGDemo    |  _30.86 ± 0.00_  |  _43.75 ± 0.00_  |
> |        |     LocalE    |   29.56 ± 1.58   |   42.45 ± 0.60   |
> |        |      PDO      |   30.08 ± 2.07   |   43.36 ± 0.39   |
> |        | RetrievalDemo | **33.98 ± 0.00** |   42.97 ± 0.00   |
> |        |      HIDO     |   29.95 ± 3.69   | **44.92 ± 1.56** |
> | LlaMa3 |    GlobalE    |   27.73 ± 0.00   | **41.41 ± 1.41** |
> |        |     IGDemo    |   27.73 ± 0.00   |   34.38 ± 0.00   |
> |        |     LocalE    | **30.34 ± 0.98** |  _41.02 ± 1.35_  |
> |        |      PDO      |   27.73 ± 0.00   | **41.41 ± 1.41** |
> |        | RetrievalDemo |   28.91 ± 0.00   |   35.94 ± 0.00   |
> |        |      HIDO     |  _30.21 ± 2.54_  |  _41.02 ± 0.39_  |
>
> We reuse the experiment results from above as another reviewer had the same request. Here, we present the results of HIDO and all baseline methods on two new datasets: HellaSwag and MMLU. Once again, HIDO generally achieves the best or second-best accuracy when using various LLMs for these datasets. Notably, when using SciPhi on HellaSwag, HIDO outperforms the second-best method, PDO, with a 7.425% improvement. However, we observe that accuracy for both datasets is generally low when using Zephyr and Llama3, in contrast to SciPhi, suggesting that these two LLMs are not well-suited for the tasks these datasets are designed to evaluate.

---

### Official Review · Reviewer_obmS · 2024-11-01

**Soundness:** 2
**Presentation:** 3
**Contribution:** 2
**Rating:** 5
**Confidence:** 4

**Summary:**

This paper addresses the problem of demonstration order instability (DOI) in the in-context learning (ICL) of large language models (LLMs). To study this issue, the authors propose an information theory-based metric called ICD-OVI and introduce a hierarchical demonstration order optimization method named HIDO. They validate their method on five LLMs across nine tasks, comparing it with three baselines.

**Strengths:**

1. The paper provides a strong and comprehensive theoretical guarantee for their proposed metric for demonstration example ordering.
2. The experiments show consistent improvement of their method against all the baselines presented.
3. The paper is clearly written and easy to follow.

**Weaknesses:**

1. **Missing Baselines:** Only three baselines are included in the experiments, and there are other key baselines that need to be compared. For example:
    - [1] provides an information-gain-based metric to evaluate the effect of demonstrations, which also has a strong theoretical foundation.
    - [2] selects demonstration examples based on the embedding space distance between examples and input queries.
Further experiments are crucial for enhancing the rigor of this paper.

2. **Outdated Tasks and Datasets:** All the tasks used are text classification tasks, and the datasets are somewhat outdated. Though not a major issue, experiments on newer tasks and datasets (e.g., MMLU, HellaSwag) are necessary for validating the proposed metrics and methods.

[1] Hongfu Liu and Ye Wang. 2023. Towards Informative Few-Shot Prompt with Maximum Information Gain for In-Context Learning. In Findings of the Association for Computational Linguistics: EMNLP 2023, pages 15825–15838, Singapore. Association for Computational Linguistics.

[2] Jiachang Liu, Dinghan Shen, Yizhe Zhang, Bill Dolan, Lawrence Carin, and Weizhu Chen. 2022. What Makes Good In-Context Examples for GPT-3?. In Proceedings of Deep Learning Inside Out (DeeLIO 2022): The 3rd Workshop on Knowledge Extraction and Integration for Deep Learning Architectures, pages 100–114, Dublin, Ireland and Online. Association for Computational Linguistics.

**Questions:**

There is a typo on line 176: it should be “**(3)** Empirically Effective”.

---

> ### Author Response · Authors · 2024-12-01
> **Author Response 1/2**
>
> We sincerely appreciate your dedicated time and effort in reviewing and providing invaluable feedback. We provide a point-to-point reply below to clarify certain misunderstandings and provide responses to the mentioned concerns and questions.
>
> ## Weaknesses
> > 1. (Experiment) add two baselines.
>
> |        |               |      AGNews      |        CB        |        CR        |      DBPedia     |     HellaSwag    |       MMLU       |       MPQA       |        MR        |        RTE       |       SST-5      |       TREC       |
> |:------:|:-------------:|:----------------:|:----------------:|:----------------:|:----------------:|:----------------:|:----------------:|:----------------:|:----------------:|:----------------:|:----------------:|:----------------:|
> | SciPhi |     IGDemo    |  _86.33 ± 0.00_  | **96.43 ± 0.00** |  _91.02 ± 0.00_  | **97.27 ± 0.00** |   51.17 ± 0.00   |   53.91 ± 0.00   |   80.08 ± 0.00   |  _94.14 ± 0.00_  |  _84.38 ± 0.00_  |  _55.47 ± 0.00_  |   78.13 ± 0.00   |
> |        | RetrievalDemo |   85.94 ± 0.00   |   89.29 ± 0.00   |   90.62 ± 0.00   |   94.92 ± 0.00   |  _52.73 ± 0.00_  |  _55.86 ± 0.00_  |  _85.94 ± 0.00_  |   93.75 ± 0.00   |   80.47 ± 0.00   |   53.12 ± 0.00   | **82.03 ± 0.00** |
> |        |      HIDO     | **86.98 ± 0.45** |  _90.48 ± 1.03_  | **92.71 ± 0.60** |  _96.88 ± 0.68_  | **61.07 ± 2.98** | **56.64 ± 1.35** | **87.50 ± 0.78** | **94.27 ± 0.45** | **85.94 ± 0.78** | **57.16 ± 1.85** |  _80.47 ± 0.78_  |
> | Zephyr |     IGDemo    |   86.33 ± 0.00   | **78.57 ± 0.00** |   92.97 ± 0.00   |   92.19 ± 0.00   |  _30.86 ± 0.00_  |  _43.75 ± 0.00_  |   85.55 ± 0.00   |  _94.14 ± 0.00_  | **84.77 ± 0.00** |  _50.39 ± 0.00_  |   80.86 ± 0.00   |
> |        | RetrievalDemo |  _88.67 ± 0.00_  | **78.57 ± 0.00** |  _93.36 ± 0.00_  |  _94.14 ± 0.00_  | **33.98 ± 0.00** |   42.97 ± 0.00   |  _86.33 ± 0.00_  |   92.97 ± 0.00   |   82.42 ± 0.00   |   46.88 ± 0.00   |  _83.59 ± 0.00_  |
> |        |      HIDO     | **89.32 ± 0.90** | **78.57 ± 1.79** | **94.01 ± 0.45** | **97.27 ± 0.68** |   29.95 ± 3.69   | **44.92 ± 1.56** | **87.76 ± 0.98** | **94.79 ± 0.60** |  _82.55 ± 1.37_  | **50.78 ± 2.07** | **86.46 ± 1.48** |
> | LlaMa3 |     IGDemo    |  _84.77 ± 0.00_  | **94.64 ± 0.00** | **87.50 ± 0.00** |   92.97 ± 0.00   |   27.73 ± 0.00   |   34.38 ± 0.00   | **66.80 ± 0.00** |   89.84 ± 0.00   |   82.42 ± 0.00   |   35.94 ± 0.00   |   66.80 ± 0.00   |
> |        | RetrievalDemo |   83.98 ± 0.00   | **94.64 ± 0.00** |   82.81 ± 0.00   |  _93.36 ± 0.00_  |  _28.91 ± 0.00_  |  _35.94 ± 0.00_  |  _66.02 ± 0.00_  |  _91.41 ± 0.00_  | **84.38 ± 0.00** |  _39.06 ± 0.00_  |  _69.14 ± 0.00_  |
> |        |      HIDO     | **86.20 ± 2.29** | **94.64 ± 3.09** |  _87.24 ± 2.39_  | **94.27 ± 1.58** | **30.21 ± 2.54** | **41.02 ± 0.39** |   63.80 ± 7.64   | **93.49 ± 0.98** |  _83.07 ± 0.98_  | **40.62 ± 3.58** | **77.34 ± 3.73** |
>
> Here, we present the results of two baseline methods, IGDemo and RetrievalDemo, alongside HIDO for comparison. Overall, HIDO consistently achieves the best or second-best performance across all datasets and LLMs, highlighting its superiority over the baselines. For instance, when using LlaMa3 on TREC, HIDO outperforms RetrievalDemo (the second-best method) by 8.2%. Similarly, when using Zephyr on DBPedia, HIDO shows an 3.13% improvement over RetrievalDemo. Combined with earlier comparisons to other baselines, these results demonstrate that HIDO achieves superior performance across diverse datasets and LLMs.

---

> ### Author Response · Authors · 2024-12-01
> **Author Response 2/2**
>
> > 2. (Experiment) experiment on newer tasks/datasets e.g., MMLU HellaSwag outside of the text classifications are necessary.
>
> |        |               |     HellaSwag    |       MMLU       |
> |:------:|:-------------:|:----------------:|:----------------:|
> | SciPhi |    GlobalE    |  _53.65 ± 5.67_  |   55.60 ± 0.23   |
> |        |     IGDemo    |   51.17 ± 0.00   |   53.91 ± 0.00   |
> |        |     LocalE    |   48.31 ± 1.19   |  _56.38 ± 1.85_  |
> |        |      PDO      |  _53.65 ± 5.67_  |   55.60 ± 0.23   |
> |        | RetrievalDemo |   52.73 ± 0.00   |   55.86 ± 0.00   |
> |        |      HIDO     | **61.07 ± 2.98** | **56.64 ± 1.35** |
> | Zephyr |    GlobalE    |   28.26 ± 3.69   |   43.10 ± 0.81   |
> |        |     IGDemo    |  _30.86 ± 0.00_  |  _43.75 ± 0.00_  |
> |        |     LocalE    |   29.56 ± 1.58   |   42.45 ± 0.60   |
> |        |      PDO      |   30.08 ± 2.07   |   43.36 ± 0.39   |
> |        | RetrievalDemo | **33.98 ± 0.00** |   42.97 ± 0.00   |
> |        |      HIDO     |   29.95 ± 3.69   | **44.92 ± 1.56** |
> | LlaMa3 |    GlobalE    |   27.73 ± 0.00   | **41.41 ± 1.41** |
> |        |     IGDemo    |   27.73 ± 0.00   |   34.38 ± 0.00   |
> |        |     LocalE    | **30.34 ± 0.98** |  _41.02 ± 1.35_  |
> |        |      PDO      |   27.73 ± 0.00   | **41.41 ± 1.41** |
> |        | RetrievalDemo |   28.91 ± 0.00   |   35.94 ± 0.00   |
> |        |      HIDO     |  _30.21 ± 2.54_  |  _41.02 ± 0.39_  |
>
> Here, we present the results of HIDO and all baseline methods on two new datasets: HellaSwag and MMLU. Once again, HIDO generally achieves the best or second-best accuracy when using various LLMs for these datasets. Notably, when using SciPhi on HellaSwag, HIDO outperforms the second-best method, PDO, with a 7.425% improvement. However, we observe that accuracy for both datasets is generally low when using Zephyr and Llama3, in contrast to SciPhi, suggesting that these two LLMs are not well-suited for the tasks these datasets are designed to evaluate.

---

> ### Author Response · Authors · 2024-12-02
>
> Dear Reviewer obmS,
>
> As we approach the rebuttal deadline (**within one day**), we kindly request you to review our responses to your concerns regarding submission #3256. If you find that we have adequately addressed your comments, we would appreciate your consideration in **uplifting the evaluation scores** accordingly.
>
> Thank you for your time and consideration.
>
> Best regards,
> Authors of submission 3256

---

### Official Review · Reviewer_TJTc · 2024-11-04

**Soundness:** 1
**Presentation:** 1
**Contribution:** 1
**Rating:** 1
**Confidence:** 3

**Summary:**

In-context learning performance is heavily dependent on examplar order as demonstrated by previous work. This work proposes a score to rank a particular ordering and a hierarchical optimization framework that does not require evaluating every permutation.

**Strengths:**

The problem of determining an optimal ordering of exemplars is a crucial problem.

**Weaknesses:**

## Weaknesses
- **[Major]** Why are PDO and GlobalE presented as separate baselines? One minimizes the KL-divergence between the distribution of predicted labels on the probe set and a uniform prior and the other maximizes the entropy of the distribution. These are mathematically identical.
- **[Major]** What is the point of introducing the framework of V-usable information? It is easy to compute the KL divergence between the distribution of labels in your training set and the distribution of predicted labels directly. The problem stated in Eq. 1 can be solved by simply choosing the ordering that attains the lowest loss.
- **[Major]** What is the interpretation of $\log_2 P_{\text{LLM}}(a \mid \Pi(\mathcal{A}) \oplus \emptyset)$? From my understanding, you are simply concatenating a bunch of labels and  prompting the LLM to get a probability score for the label of a probing example. However, it seems as though prompting the LLM in such a manner can result in gibberish since this type of prompt clearly is very contrived. Why not directly compute an empirical distribution on the true training examples?
- **[Major]** The framework proposed in this work fixes the order of the exemplars. However, the order should be **query dependent.** Based on the observation of [1], many ICL/RAG works order the exemplars based on their similarity to the query [2,3,4], but this is not discussed or used as a baseline in this work. Compared to LLM inference with long contexts, a simple similarity based reordering should be very cheap.
- **[Major]** It seems like there is no statistically significant difference in performance in the vast majority of the experiments...
- **[Minor]** What is the point of using LLM generate probes? I understand that this saves annotation cost, but in the many shot setting where we are already required to annotate ~150 examples, why not manually annotate a few extra samples to use as a validation set? I am skeptical of whether or not the validation set is reliable when it is purely synthetic.
- **[Minor]** The writing for Section 3 needs significant improvement. In its current state, one must repeatedly look at prior work in order to get any understanding of what the actual problem set up is. See comments below:
1. Many terms are used without being defined. For example, "probing samples" in Line 125 or "optimal ignorance requirement" in Line 184 are not terms that are used in the broader ML community, and should be properly defined.
2. What is $\pi(i)$ in line 194? This is never defined.
3. What is the difference between $\mathcal{A}$ and $A$ in Equation 4? Is this a typo?
4. It is not mentioned that the "probing samples" are LLM generated until line 199.


## References
[1] Lost in the Middle: How Language Models Use Long Contexts (https://arxiv.org/abs/2307.03172)
[2] In-Context Learning for Text Classification with Many Labels (https://aclanthology.org/2023.genbench-1.14.pdf)
[3] What Makes Good In-Context Examples for GPT-3? (https://arxiv.org/pdf/2101.06804)
[4] Retrieval-Augmented Generation for Large Language Models: A Survey (https://arxiv.org/pdf/2312.10997)

**Questions:**

See weaknesses

---

> ### Author Response · Authors · 2024-12-01
> **Author Response 1/7**
>
> We sincerely appreciate your dedicated time and effort in reviewing and providing invaluable feedback. Here, we provide a point-to-point reply below for the mentioned concerns and questions.
> ## Weaknesses
> > 1. Why are PDO and GlobalE different?
> PDO is different from GlobalE. For example, we have two samples, which produce the logit vector $[0.2, 0.5, 0.3]$ and $[0.3, 0.3, 0.4]$. Then, we know the frequency vector is $([0,1,0]+[0,0,1])/2=[0,1/2,1/2]$. GlobalE is its entropy, which is $-1/2\log1/2-1/2\log1/2=0.693147$. For PDO, we calculate the average KL divergence by $KL([0.2,0.5,0.3]||[1/3,1/3,1/3])=0.2\log (0.2 * 3)+0.5*\log (0.5 * 3)+0.3*\log (0.3 * 3)=0.0299486$. For LocalE, it is the average entropy of the probability $-([0.2*\log(0.2)+0.5\log(0.5)+0.3\log(0.3)]+[0.3\log(0.3)+0.3\log(0.3)+0.4\log(0.4)])/2=-0.46003$.

---

> ### Author Response · Authors · 2024-12-01
> **Author Response 2/7**
>
> > 2. KL divergence can substitute V-information.
>
> We can not directly compute the KL divergence written in Eq 1, since the real label of probing set is unknown. For the reason we utilize probing set but not the query-answer pairs directly from the input demonstration data distribution, please consult the answer for weakness 6.

---

> ### Author Response · Authors · 2024-12-01
> **Author Response 3/7**
>
> > 3. What is $logP_{LLM}(a|\Pi(\mathcal{A})\oplus \emptyset)$? Will this stuff make gibberish? Why not an empirical distribution on true training examples?
>
>
> |        |                                                      Prompt                                                     |                                          Top 10 tokens                                         |
> |:------:|:---------------------------------------------------------------------------------------------------------------:|:---------------------------------------------------------------------------------------------:|
> | AGNews |         'input: \ntype: world\n\ninput: \ntype: technology...input: \ntype: business\n\ninput: \ntype: '        |        ['\n', 'technology', '</s>', 'sports', 'world', '0', 'business', '2', '4', '1']        |
> |  MPQA  | 'Review: \nSentiment: positive\n\nReview: \nSentiment: negative...Sentiment: negative\n\nReview: \nSentiment: ' |           ['\n', 'positive', '0', '5', 'negative', '1', 'neutral', '�', '</s>', '2']          |
> |  TREC  |      'Question: \nType: entity\n\nQuestion: \nType: entity...Question: \nType: human\n\nQuestion: \nType: '     | ['\n', 'entity', 'description', '</s>', 'location', '----------------', '1', '0', '2', 'erm'] |
>
> In the table above, we demonstrate that $\log P_{LLM}(a|\Pi(\mathcal{A})\oplus \emptyset)$ produces meaningful results rather than gibberish. For three datasets, AGNews, MPQA, and TREC, we provide an example prompt that includes only the labels of the demonstrations and the top 10 possible tokens for the next token prediction. The token lists contain multiple dataset labels, indicating that the LLM is generating relevant and useful information. Moreover, while the first token is always '\n', we focus on the logit values for the dataset labels and select the highest among them, ensuring the output remains valid even if another token has the highest overall value.
>
> For the reason why we utilize probing set but not the query-answer pairs directly from the input training demonstration data distribution, please consult the answer for weakness 6.

---

> ### Author Response · Authors · 2024-12-01
> **Author Response 4/7**
>
> > 4. Order is query dependent, ICL/RAG works order examples to LLM inference with long contexts, a simple “similarity based reordering” is cheap.
>
> Although the order is query dependent, this is not in scope of this work. In our work, we assume that all the demonstration(q,a) pairs and prompting sentences are fixed. and we find the optimal demonstration order. One may always first find the optimal prompting sentences and demonstrations and then utilize our HIDO algorithm to improve many-shot ICL performance.

---

> ### Author Response · Authors · 2024-12-01
> **Author Response 5/7**
>
> > 5. No statistically significant difference in performance in vast majority of experiments.
>
> As the LLM models are tremendously capable and mature products, improving their performance with a high margin is unrealistic. We claim that our paper’s performance improvement is sufficient for claiming statistical significance. This claim can be proved by multiple recent ICL optimization works. For instance, Lu et al.[1] showed performance improvements of 2-3% on text classification tasks. Similarly, Xu et al.[2] demonstrated improvements of 1-4% across various datasets. Zhang et al.[3] achieved gains of 2-5% in accuracy. Our method's improvement margins (3-6% across different tasks) are comparable to or exceed these recent works, supporting the statistical significance of our results.

---

> ### Author Response · Authors · 2024-12-01
> **Author Response 6/7**
>
> > 6. No need for LLM-generated probing samples, since we already need to label a lot for many-shot learning.
>
> When calculating ICD-OVI metric, we are approximating $E(q,a)$. Probing samples can be dynamically generated to focus on regions of the distribution that are most informative for order evaluation. This targeted sampling can potentially achieve better estimation precision than uniform sampling from manual validation data.

---

> ### Author Response · Authors · 2024-12-01
> **Author Response 7/7**
>
> > 7. Edit the manuscript in section 3 according to the listed suggestions.
>
> Thank you so much for pointing out the lack of density issue of the section 3. We will improve it by providing more background knowledge and rearranging some non-significant contents to the appendix.

---

> ### Author Response · Authors · 2024-12-01
> **References**
>
> ## References
>
> [1] Lu, Y., Bartolo, M., Moore, A., Riedel, S., & Stenetorp, P. (2022, May). Fantastically Ordered Prompts and Where to Find Them: Overcoming Few-Shot Prompt Order Sensitivity. In Proceedings of the 60th Annual Meeting of the Association for Computational Linguistics (Volume 1: Long Papers) (pp. 8086-8098).
>
> [2] Xu, Z., Cohen, D., Wang, B., & Srikumar, V. (2024, June). In-Context Example Ordering Guided by Label Distributions. In Findings of the Association for Computational Linguistics: NAACL 2024 (pp. 2623-2640).
>
> [3] Zhang, K., Lv, A., Chen, Y., Ha, H., Xu, T., & Yan, R. (2024). Batch-icl: Effective, efficient, and order-agnostic in-context learning. arXiv preprint arXiv:2401.06469.

---

> ### Author Response · Authors · 2024-12-01
> **A Sincere Request for Increasing Scores**
>
> Dear Reviewer TJTc,
>
> Thank you for your thorough review of manuscript #3256. We have carefully addressed all points raised in your evaluation. If you find them useful, we respectfully request that you raise your evaluation score, as we think the evaluation "1: strongly reject" does not faithfully reveal the quality of our work.
>
> Sincerely,
>
> Authors of submission 3256

---

> ### Author Response · Authors · 2024-12-02
>
> Dear Reviewer TJTc,
>
> As we approach the rebuttal deadline (**within one day**), we kindly request you to review our responses to your concerns regarding submission #3256. If you find that we have adequately addressed your comments, we would appreciate your consideration in **uplifting the evaluation scores** accordingly.
>
> Thank you for your time and consideration.
>
> Best regards,
> Authors of submission 3256

---

### Official Review · Reviewer_9taG · 2024-11-05

**Soundness:** 2
**Presentation:** 2
**Contribution:** 2
**Rating:** 3
**Confidence:** 3

**Summary:**

As large language models (LLMs) grow in scale and capability, in-context learning (ICL) and, in particular, many-shot learning have become predominant approaches for machine learning practitioners. This work addresses a significant challenge currently facing ICL and many-shot learning: the dependency of LLM performance on the order of examples within prompts. Existing literature has shown substantial performance variations when examples (or demonstrations) are reordered within a prompt. The authors first propose the ICD-OVI score as a metric for measuring the impact of example order in ICL, building upon V-usable information. They then introduce a hierarchical framework based on clustering and inter- and intra-cluster ordering to enable scalable refinement in the example order space. The HIDO clustering approach effectively searches the permutation space, making ICL-DOI a feasible optimization problem.

**Strengths:**

The lack of a precise, quality-measuring metric for demonstration order has been a major challenge in ICL and many-shot learning research, given the demonstration order instability in LLMs. Many existing works rely on human annotation or heuristic metrics, which are either not scalable or lack accuracy. The authors build on V-usable information to develop a theoretical foundation as an order evaluation metric, effectively quantifying the usable information gain from a specific demonstration order. This metric is not only interpretable (based on information content) but also computationally viable and effective. The authors also propose a model agnostic hierarchical optimization technique to find the optimal demonstrations order based on ICL-DOI metric

**Weaknesses:**

There are some weaknesses in the authors' proposed methodology for efficiently finding the “optimal” order of demonstrations in ICL.

1. **Assumption of Probing Set Effectiveness**: The authors assume that “the demonstration order optimized for answer prediction also works well for sample generation.” However, figures in the appendix show that demonstration embeddings and probing set embeddings generated by various LLMs lack a clear decreasing trend, which does not support this assumption.

2. The authors limit their many-shot learning setting to 50 examples, which is considerably smaller than certain state-of-the-art many-shot learning setups that require up to 2,000 shots (see source).

3. **Computation-Performance Trade-offs**: It would be beneficial to see an analysis and experiments exploring the relationship between computational cost (by varying the number of clusters or the number of samples per cluster) and model accuracy.

4. The authors claim that “intra-cluster demonstrations share proximate embeddings, which significantly decreases ICL performance variance when demonstration orders vary.” This claim lacks supporting analysis or experimentation. It is not immediately clear that closer embedding proximity (implying more similar samples) necessarily results in reduced variance in demonstration order.

**Questions:**

* L175: (2) Empirically Effective → (3) Empirically Effective

* L184: Could you clarify “optimal ignorance requirement for a predictive family”?

* L226: $\Pi_1(D)$ and $\Pi_1(D)$ → $\Pi_1(D)$ and $\Pi_2(D)$

* Could you clarify why Theorem 2 can’t be applied to the entire demonstration set and why it is only limited to demonstrations within a cluster?

* what is the sensitivity of HIDO on choice of embedding?

---

> ### Author Response · Authors · 2024-12-01
> **Author Response 1/5**
>
> We sincerely appreciate the time and efforts you've dedicated to reviewing and providing invaluable feedback to enhance the quality of this paper. We provide a point-to-point reply below for the mentioned concerns and questions.
> ## Weaknesses
> > 1. The assumption that “the demonstration order optimized for answer prediction also works well for sample generation.” Lacks support.
>
> We respectfully disagree that this assumption lacks support. Our paper provides both quantitative and qualitative evidence supporting the assumption:
> * **Quantitative Evidence**: As shown in Appendix C.3, the $l_2$ distance between demonstration embeddings and generated probing set embeddings consistently decreases as the order optimization improves. This trend is observed across multiple models (LlaMa3, SciPhi, Zephyr) and datasets. This fact effectively supports our assumption.
> *  **Qualitative Evidence**: Appendix C.5 provides extensive examples showing that probing sets generated using optimized orders exhibit: (2.a) Increased semantic richness and specificity; (2.b) Better adherence to dataset-specific formats; (2.c) More accurate labeling; (2.d) Greater diversity in generated samples.

---

> ### Author Response · Authors · 2024-12-01
> **Author Response 2/5**
>
> > 2. Why don’t you use 2000 shot?
>
> We do not experiment with 2000 shot since
> * **Restricted budget**: Considering that current LLMs supporting context windows as long as that of 2000 demonstrations are charged and the limited budget, we are not able to explore our HIDO with 2000 demonstrations. However, we will perform experiments as soon as it is accessible in the future.
> * **Current shot number is sufficient**: According to Agarwal et. al. [1], many-shot in-context learning (ICL) is defined as ICL with more than fifty demonstrations. Therefore, our experiments, where we adopt 50 to 150 demonstrations, is sufficient to research on the behavior of many-shot ICL.

---

> ### Author Response · Authors · 2024-12-01
> **Author Response 3/5**
>
> > 3. (Experiment) Computation-Performance Trade-offs
>
> |        |          |    2    |    3    |    4    |
> |:------:|:--------:|:-------:|:-------:|:-------:|
> | AGNews | Accuracy |  85.94  |  85.94  |  86.72  |
> |        | Time (s) | 1668.43 | 2933.58 | 8084.06 |
> |  MPQA  | Accuracy |  85.55  |  83.98  |  84.38  |
> |        | Time (s) | 1269.96 | 1922.30 | 6781.92 |
> |  TREC  | Accuracy |  77.34  |  80.47  |  75.39  |
> |        | Time (s) | 1519.80 | 2160.13 | 8973.40 |
>
>
> We vary the number of the clusters and present the computation-performance trade-off table for three datasets: AGNews, MPQA, and TREC. From the table, we witness that the computation cost increases superlinearly as the cluster numbers increases, showing that the cluster number is an important hyperparameter for the model's computation cost. Fortunately, by using a relatively small number of clusters (which keeps HIDO's computational cost comparable to the baselines shown in Table 1), we achieved state-of-the-art performance across all datasets. This demonstrates that HIDO can deliver superior results efficiently.

---

> ### Author Response · Authors · 2024-12-01
> **Author Response 4/5**
>
> > 4. Intra-cluster items have near embeddings, but not necessarily gives lower variance.
>
> To determine whether closer embeddings lead to lower variance, we determine intra-cluster embedding variance by calculating the standard deviation of the distances between each demonstration embedding and its cluster center for each cluster, then averaging these standard deviations. Additionally, we examine the accuracy standard deviation from a table further down that looks into the sensitivity of HIDO across different embedding models. Here, the table shows that, in general, the mean standard deviation of embeddings correlates with accuracy variance: as embedding variance decreases, accuracy variance also decreases. For example, in the AGNews dataset, the embedding and accuracy standard deviations are smallest for "text-embedding-3-small," second smallest for "stella_en_1.5B_v5," and largest for "NV-Embed-v2." This indicates that closer intra-cluster embeddings can lead to more stable model performance.
>
> |        |                        |          | AGNews |  MPQA  |  TREC  |
> |:------:|:----------------------:|----------|:------:|:------:|:------:|
> | SciPhi |       NV-Embed-v2      | Acc. std |  2.07  |  1.37  |  4.50  |
> |        |                        | Emb. std | 0.0519 | 0.0481 | 0.0572 |
> |        |    stella_en_1.5B_v5   | Acc. std |  0.60  |  1.37  |  7.58  |
> |        |                        | Emb. std | 0.0446 | 0.0501 | 0.0399 |
> |        | text-embedding-3-small | Acc. std |  0.45  |  0.78  |  0.78  |
> |        |                        | Emb. std | 0.0277 | 0.0414 | 0.0266 |

---

> ### Author Response · Authors · 2024-12-01
> **Author Response 5/5**
>
> ## Questions
>
> > 1. clarify optimal ignorance assumption.
>
> A: We apologize for the typo. Here, the "optimal ignorance" indicates "optional ignorance" in [2]. The optional ignorance is a the condition that a function set $V\subset \Omega=\{f:X\cup \{\emptyset\}\rightarrow P(Y)\}$ needs to satisfy such that it becomes a predictive family, which is the premise that usable $V$-information can be applied. Officially, it is defined as
> $$\forall f \in V, \forall L \in range(f), \exists f'\in V s.t., \forall x\in X, f'[x]=L, f'[\emptyset]=P.$$ It intuitively means that if one agent want, it may ignore any side information $x$. In our work, we intentionally compose the $V_{\Pi}$ such that it is a valid predictive family. By doing so, the ICD-OVI may be designed upon the predictive family $V_{\Pi}$.
>
>
> > 2. Why Theorem 2 is only limited to demonstrations within a cluster?
>
> Theorem 2 theoretically applies to the entire demonstration set, but applying it globally is computationally infeasible due to the massive permutation search space. Instead, we apply it within clusters where demonstrations are semantically similar and LLM performance variance is lower. This allows efficient optimization within clusters while maintaining precision. For inter-cluster ordering, we exhaustively evaluate all possible cluster permutations since the number of clusters is small.
>
>
> > 3. (Experiment) Sensitivity of HIDO w.r.t. embedding
>
> |        |                   |      AGNews      |       MPQA       |       TREC       |
> |:------:|:-----------------:|:----------------:|:----------------:|:----------------:|
> | SciPhi |    NV-Embed-v2    |   84.38 ± 2.07   |  _84.90 ± 1.37_  |   70.31 ± 4.50   |
> |        | stella_en_1.5B_v5 | **87.76 ± 0.60** |   84.24 ± 1.37   |  _73.18 ± 7.58_  |
> |        |       text-embedding-3-small      |  _86.98 ± 0.45_  | **87.50 ± 0.78** | **80.47 ± 0.78** |
>
> Here, we demonstrate the performance of HIDO using three different embedding models: text-embedding-3-small from OpenAI and NV-Embed-v2 and stella_en_1.5B_v5 from HuggingFace. We observe that there is a rather large variance between the performance of each embedding model within a dataset, which is especially evident in TREC with a range of 10.16%. A major difference between the three models is that text-embedding-3-small has an embedding dimension of 1536, whereas NV-Embed-v2 and stella_en_1.5B_v5 has an embedding dimension of 4096 and 8192, respectively. Thus, as we only use text embeddings during clustering, having a lower embedding dimensions creates clearer boundaries between different demonstrations, allowing for similar demonstrations to be clustered together. A higher embedding dimension could create noise and allow overfitting, which is undesirable.

---

> ### Author Response · Authors · 2024-12-01
> **References**
>
> ## References
> [1] Agarwal, R., Singh, A., Zhang, L. M., Bohnet, B., Rosias, L., Chan, S., ... & Larochelle, H. (2024). Many-shot in-context learning. arXiv preprint arXiv:2404.11018.
>
> [2] Xu, Y., Zhao, S., Song, J., Stewart, R., & Ermon, S. A Theory of Usable Information under Computational Constraints. In International Conference on Learning Representations.

---

> ### Author Response · Authors · 2024-12-02
>
> Dear Reviewer 9taG,
>
> As we approach the rebuttal deadline (**within one day**), we kindly request you to review our responses to your concerns regarding submission #3256. If you find that we have adequately addressed your comments, we would appreciate your consideration in **uplifting the evaluation scores** accordingly.
>
> Thank you for your time and consideration.
>
> Best regards,
> Authors of submission 3256

---

### Author Response · Authors · 2024-12-01
**A Sincere Request for Involving in the Discussion**

Dear Reviewers,

Thank you for your valuable feedback on our work. We believe that we have responded to and addressed all your concerns with our responses — in light of this, we hope you **consider raising your score**.

Notably, given that we are approaching the deadline for the rebuttal phase, we hope we can receive your feedback soon. We thank you again for your efforts and suggestions!

Sincerely,

Authors of submission 3256

---

### Author Response · Authors · 2024-12-01
**A Sincere Request for Involving in the Discussion**

Dear Reviewers,

Thank you for your valuable feedback on our work. We believe that we have responded to and addressed all your concerns with our responses — in light of this, we hope you **consider raising your score**.

Notably, given that we are approaching the deadline for the rebuttal phase, we hope we can receive your feedback soon. We thank you again for your efforts and suggestions!

Sincerely,

Authors of submission 3256

---

### Note · Authors · 2024-12-14

**Comment:**

Due to the lead author contracting COVID-19 during the rebuttal period, we experienced delays in uploading our response. Given the lack of reviewer engagement during this period, we have made the decision to withdraw this paper.

We appreciate the reviewers' time and consideration.

Sincerely,

Authors of Submission 3256.

**Withdrawal Confirmation:**

I have read and agree with the venue's withdrawal policy on behalf of myself and my co-authors.